# Specificity, synergy, and mechanisms of splice-modifying drugs

Yuma Ishigami [1,5], Mandy S. Wong[1,3,5], Carlos Martí-Gómez[1], Andalus Ayaz[1], Mahdi Kooshkbaghi[1,4], Sonya M. Hanson [2], David M. McCandlish [1], Adrian R. Krainer [1,6] & Justin B. Kinney [1,6]

Drugs that target pre-mRNA splicing hold great therapeutic potential, but the quantitative understanding of how these drugs work is limited. Here we introduce mechanistically interpretable quantitative models for the sequence-specific and concentration-dependent behavior of splice-modifying drugs. Using massively parallel splicing assays, RNA-seq experiments, and precision dose-response curves, we obtain quantitative models for two small-molecule drugs, risdiplam and branaplam, developed for treating spinal muscular atrophy. The results quantitatively characterize the specificities of risdiplam and branaplam for 5' splice site sequences, suggest that branaplam recognizes 5' splice sites via two distinct interaction modes, and contradict the prevailing two-site hypothesis for risdiplam activity at *SMN2* exon 7. The results also show that anomalous single-drug cooperativity, as well as multi-drug synergy, are widespread among small-molecule drugs and antisense-oligonucleotide drugs that promote exon inclusion. Our quantitative models thus clarify the mechanisms of existing treatments and provide a basis for the rational development of new therapies.

Alternative pre-mRNA splicing has become a major focus of drug development[1–10]. Three splice-modifying drugs—nusinersen, risdiplam, and branaplam—have been developed to treat spinal muscular atrophy (nusinersen and risdiplam have been approved for clinical use; branaplam has been withdrawn). All three drugs function by promoting the inclusion of *SMN2* exon 7. Nusinersen[11–13] is an antisense oligonucleotide (ASO) that binds to a complementary site in intron 7 of *SMN2* pre-mRNA, thereby blocking the binding of the splicing repressors hnRNP A1/A2 and promoting exon 7 inclusion. Unlike nusinersen, risdiplam[14–16] (Fig. 1A) and branaplam[17–19] (Fig. 1B) are small molecules, and the mechanisms by which they promote *SMN2* exon 7 inclusion are not well understood.

Biochemical studies[18,20] have shown that both risdiplam and branaplam bind to and stabilize the complex formed by the U1 snRNP and the 5' splice site (5'ss) of *SMN2* exon 7. However, it is unclear why risdiplam and branaplam are so specific for *SMN2* exon 7, as opposed

to other exons in the human genome. It is also unclear why branaplam is less specific than risdiplam for *SMN2* exon 7[18]. Two independent studies[18,21] have proposed that risdiplam (but not branaplam) further promotes *SMN2* exon 7 inclusion by binding to a second RNA site within exon 7, and that the presence of this second site substantially increases the specificity of risdiplam relative to branaplam. This two-site hypothesis has become the prevailing explanation for risdiplam's specificity[1,20,22–31], but the mechanism by which risdiplam recognizes the second putative RNA site remains unclear, as does the quantitative influence that the second putative RNA site has on *SMN2* exon 7 inclusion.

A major obstacle to defining the specificities of risdiplam and branaplam, as well as the mechanistic basis for each drug's specificity, is the lack of mechanistically interpretable quantitative models for how splice-modifying drugs affect their targets. Biophysically interpretable models for protein-targeted drugs have existed for over a century and

[1]Cold Spring Harbor Laboratory, Cold Spring Harbor, NY 11724, USA. [2]Flatiron Institute, New York, NY 10010, USA. [3]Present address: Beam Therapeutics, Cambridge, MA 02142, USA. [4]Present address: The Estée Lauder Companies, New York, NY 10153, USA. [5]These authors contributed equally: Yuma Ishigami, Mandy S. Wong. [6]These authors jointly supervised this work: Adrian R. Krainer, Justin B. Kinney. ✉e-mail: krainer@cshl.edu; jkinney@cshl.edu

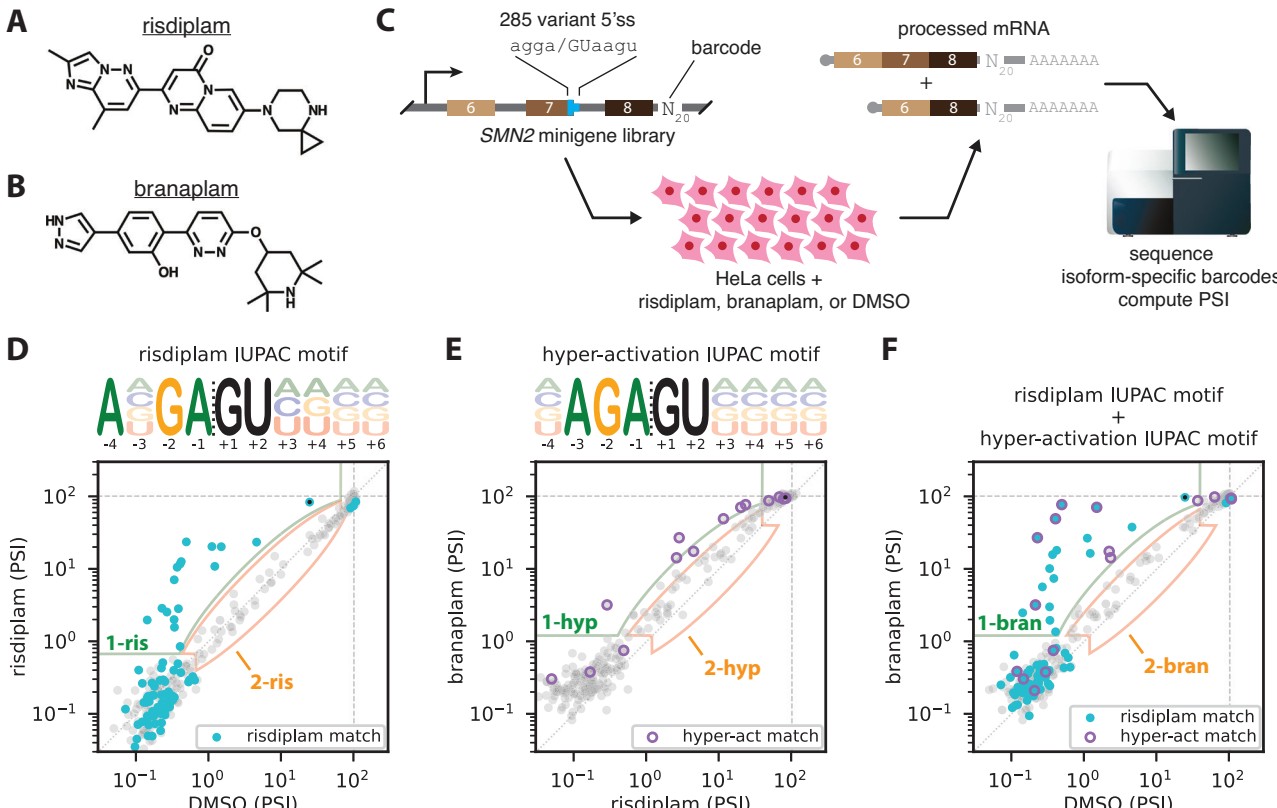

**Fig. 1 | MPSAs reveal IUPAC motifs for the 5'ss specificities of risdiplam and branaplam. A, B** Structures of (**A**) risdiplam and (**B**) branaplam. **C** MPSA performed in the context of a minigene library spanning exons 6, 7, and 8 of *SMN2*. In this library, the 5'ss of exon 20 was replaced by 285 variant 5'ss sequences of the form agga/GUaagu, where lower-case letters indicate mutagenized positions. **D–F** PSI values measured in the presence of (**D**) risdiplam vs. DMSO, (**E**) branaplam vs. risdiplam, or (**F**) branaplam vs. DMSO. Black dots, wild-type *SMN2* exon 7 5'ss (AGGA/GUAAGU). Cyan dots, 5'ss that match the risdiplam IUPAC motif. Purple

circles, 5'ss that match the hyper-activation IUPAC motif. Light green outlined areas, 5'ss classified as activated by y-axis treatment relative to x-axis treatment (class 1-ris in panel **D**, class 1-hyp in panel **E**, class 1-bran in panel **F**). Peach outlined areas, 5'ss classified as insensitive to y-axis treatment relative to x-axis treatment (class 2-ris in panel D, class 2-hyp in panel **E**, class 2-bran in panel **F**). MPSA, massively parallel splicing assay. 5'ss, 5' splice site. PSI, percent spliced in. DMSO, dimethyl sulfoxide.

are routinely used to discern drug mechanism[32–34], but the complexity of splicing prevents the direct application of these models to splice-modifying drugs. There remains the possibility, however, that relatively simple quantitative models—ones that approximate the complex behavior of the spliceosome using a small number of parameters—might be useful for understanding how splice-modifying drugs work. Indeed, such quantitative models have been proposed to describe how splicing is affected by genetic variation[35], but analogous models that describe how splicing is affected by splice-modifying drugs have yet to be developed.

Here we introduce quantitative models for the sequence-specific and concentration-dependent effects of splice-modifying drugs. We first report the results of massively parallel splicing assays[36] (MPSAs) and RNA-seq experiments performed to study the specificity of risdiplam and branaplam for variant 5'ss sequences. An exploratory analysis of these data reveals that the 5'ss specificity of risdiplam is well described by a single IUPAC motif (i.e., a qualitative motif that specifies the compatible bases at each nucleotide position[37]), but that the specificity of branaplam is not. Rather, the 5'ss specificity of branaplam is well-described by a combination of two distinct IUPAC motifs. Based on these exploratory findings, and motivated by the success of quantitative biophysical modeling approaches in transcriptional regulation[38–43] and RNA biology[35,44], we propose a quantitative biophysical model in which branaplam recognizes 5'ss sequences via two distinct interaction modes. We find that this two-interaction-mode model for branaplam explains MPSA and RNA-seq data much better than a model that assumes only one interaction mode for branaplam.

We then discuss the implications of this two-interaction-mode model in light of existing structural data.

Next we report the results of dose-response measurements for risdiplam and branaplam. The results contradict the two-site hypothesis for risdiplam activity at *SMN2* exon 7 and provide an alternative explanation for previously reported evidence offered in favor of the two-site hypothesis. Finally, we report dose-response measurements for other splice-modifying drugs, as well as for mixtures of splice-modifying drugs. The results reveal anomalous single-drug cooperativity in the dose-response behavior of multiple splice-modifying drugs, as well as widespread multi-drug synergy between splice-modifying drugs that target the same exon. Our study thus quantitatively characterizes the sequence-specificities of risdiplam and branaplam, changes the mechanistic understanding of how these drugs function, and suggests novel strategies for developing therapeutics.

## Results

### MPSAs quantify the 5'ss-dependent effects of risdiplam and branaplam

To characterize the 5'ss specificities of risdiplam and branaplam, we used MPSAs to measure the effects that risdiplam and branaplam have on variant 5'ss sequences in a fixed genetic context (Fig. 1C). We constructed a library of 285 three-exon minigenes (spanning exons 6, 7, and 8 of *SMN2*, with intron 6 truncated) in which all possible single-position and pairwise mutations were introduced at 8 positions of the exon 7 5'ss (agga/GUaagu, mutated positions in lowercase); see Supplemental Information (SI) Sec. 1.4 for details. This minigene library

was transiently transfected into HeLa cells in the presence of 100 nM risdiplam, 50 nM branaplam, or DMSO. Note that different drug concentrations were used to compensate for differences in drug potency; see SI Sec. 1.5. The abundance of minigene mRNA including exon 7 relative to total minigene mRNA was then quantified using reverse transcription and high-throughput sequencing of barcodes present in different populations of isoforms. Positive and negative controls confirmed the ability of this MPSA to precisely measure percent spliced in (PSI) over a ~300× dynamic range; PSI was limited on the low end by the use of cryptic 5′ss (Fig. S1). The resulting PSI values are plotted in Fig. 1D–F.

To study the 5′ss specificities of risdiplam and branaplam in a different gene context, we repeated these experiments using our previously reported[36] minigene library for *ELP1* exon 20, which contains nearly all 32,768 variant 5′ss of the form ANNN/GYNNNN (Fig. S2A). The resulting PSI values are plotted in Fig. S2B–D.

## MPSAs identify an IUPAC motif for 5′ss activation by risdiplam
We next carried out an exploratory analysis of the sequence specificity of risdiplam using qualitative IUPAC motifs[37,45]. Specifically, we sought to identify an IUPAC motif that describes which 5′ss in the *SMN2* minigene library were activated by risdiplam relative to DMSO. Based on PSI measurements in the presence of risdiplam or DMSO, we categorized each assayed 5′ss into one of three classes: (class 1-ris) 5′ss activated by risdiplam; (class 2-ris) 5′ss insensitive to risdiplam; or (neither class 1-ris nor class 2-ris) 5′ss for which the effect of risdiplam could not be confidently determined, due to PSI in the absence of drug being too high or PSI in the presence of drug being too low; see SI Sec. 2.2 for details. We then searched for IUPAC motifs that matched all 5′ss in class 1-ris and did not match any 5′ss in class 2-ris. Multiple IUPAC motifs met these classification criteria. One motif that met the classification criteria was ANGA/GUHDNN (Fig. 1D); we call this the "risdiplam IUPAC motif". See SI Sec. 2.3 (including Fig. S3A, B) for a discussion of other motifs that matched the classification criteria. We also observed that the 5′ss sequences that match the risdiplam IUPAC motif tended to be activated by risdiplam relative to DMSO in the *ELP1* minigene library (Fig. S2B). We conclude that the risdiplam IUPAC motif provides a plausible (though not unique) qualitative explanation for the specificity of risdiplam observed in our MPSA experiments.

## MPSAs identify a two-IUPAC-motif model for 5′ss activation by branaplam
We next carried out an exploratory analysis of the sequence specificity of branaplam. As with risdiplam, we sought to identify a qualitative IUPAC motif that describes which 5′ss in our minigene library were activated by branaplam relative to DMSO. We categorized each assayed 5′ss into one of three classes: (class 1-bran) 5′ss activated by branaplam; (class 2-bran) 5′ss insensitive to branaplam; or (neither class 1-bran nor class 2-bran) 5′ss for which the effect of branaplam could not be confidently determined. Unlike with risdiplam, however, no IUPAC motifs met these classification criteria. This finding was not sensitive to the boundaries used to define class 1-bran or class 2-bran (Fig. S4, SI Sec. 2.2). We conclude that the specificity of branaplam for 5′ss in our MPSA experiments cannot be qualitatively described by a single IUPAC motif.

We therefore considered an alternative hypothesis: that there might be an IUPAC motif that describes activation by branaplam relative to risdiplam. This hypothesis was motivated by the observations that most assayed 5′ss in the *SMN2* minigene library exhibited similar PSI in the presence of branaplam relative to risdiplam, that no 5′ss showed substantially higher PSI in the presence of risdiplam relative to branaplam, and that a few 5′ss exhibited substantially higher PSI in the presence of branaplam relative to risdiplam (Fig. 1E). We therefore categorized each assayed 5′ss into one of three classes: (class 1-hyp) 5′ss activated by branaplam relative to risdiplam; (class 2-hyp)

5′ss with similar sensitivity to branaplam and risdiplam; or (neither class 1-hyp nor class 2-hyp) 5′ss for which the effect of branaplam relative to risdiplam could not be confidently determined. Multiple IUPAC motifs met these classification criteria. One motif that met the classification criteria was NAGA/GUNNNN (Fig. 1E); we call this the "hyper-activation IUPAC motif". See Fig S3C, D for other motifs that matched the classification criteria. 5′ss that match the hyper-activation IUPAC motif also tended to be activated by branaplam relative to risdiplam in the *ELP1* minigene library (Fig. S2C). We conclude that the hyper-activation IUPAC motif provides a plausible (though not unique) qualitative explanation for the specificity of branaplam relative to risdiplam observed in our MPSA experiments.

The above result suggested that, instead of being well-described by a single IUPAC motif, the specificity of branaplam might instead be well-described by a two-IUPAC-motif model in which a 5′ss is activated by branaplam if it matches either the risdiplam IUPAC motif or the hyper-activation IUPAC motif. Consistent with the two-motif model, Fig. 1F shows that every 5′ss activated by branaplam relative to DMSO in the *SMN2* minigene library (class 1-bran) matches either the risdiplam IUPAC motif or the hyper-activation IUPAC motif, and that every 5′ss insensitive to branaplam (class 2-bran) matches neither the risdiplam IUPAC motif nor the hyper-activation IUPAC motif. Moreover, 5′ss that match the risdiplam IUPAC motif or the hyper-activation IUPAC motif tended to be activated by branaplam relative to DMSO in the *ELP1* minigene library (Fig. S2D). We conclude that the two-motif model provides a plausible qualitative explanation for the specificity of branaplam observed in our MPSA experiments.

## Allelic manifolds quantify 5′ss-dependent drug effects from RNA-seq data
We next asked whether the one-motif model for risdiplam and the two-motif model for branaplam were consistent with the effects of these two drugs on exons in the human transcriptome. To answer this question, we performed RNA-seq experiments in HeLa cells treated with risdiplam, branaplam, or DMSO. As in the MPSA experiments, different drug concentrations were used to compensate for differences in drug potency. PSI values were determined for all internal exons using the rMATS software package[46] (Fig. 2A). See SI Sec. 1.7 for details.

We then sought to quantify the 5′ss-dependent effects of risdiplam and branaplam observed in these RNA-seq data. Quantifying drug effect, however, required distinguishing the influence of 5′ss sequence from the influence of other genomic context factors, including other splice-regulatory elements. We therefore proposed a thermodynamic model (the "drug-effect model"; Fig. 2B and Fig. S5) for drug-dependent exon inclusion. The drug-effect model predicts PSI as a function of two quantities: drug effect (quantified by $E$), and context strength (quantified by $S$). The model assumes that PSI is given by 100 times the equilibrium occupancy of U1 binding to 5′ss RNA. The model also assumes that $E$ is dependent on 5′ss sequence, but not on other locus-specific factors that influence context strength. See SI Sec. 3.1 and Fig. S5 for details.

Next we determined 5′ss-specific values for drug effect $E$ via the Bayesian inference of allelic manifolds[47]. Fig. 2C illustrates the general form of the allelic manifolds we inferred. Each allelic manifold corresponded to a 5′ss of the form NNNN/GUNNNN that occurred in a sufficient number of exons in the human transcriptome. The shape of the allelic manifold was parameterized by the drug effect $E$ of the corresponding 5′ss, and the position along the allelic manifold of each two-dimensional exon-specific measurement (PSI in the presence and absence of drug) was parameterized by the context strength $S$ of that exon. The inference of these allelic manifolds thus involved the determination of 194,129 parameters: context strength values $S$ for 189,087 different exons, and drug effect values $E$ (one for risdiplam and one for branaplam) for 2521 distinct 5′ss sequences; see SI Sec. 4.1

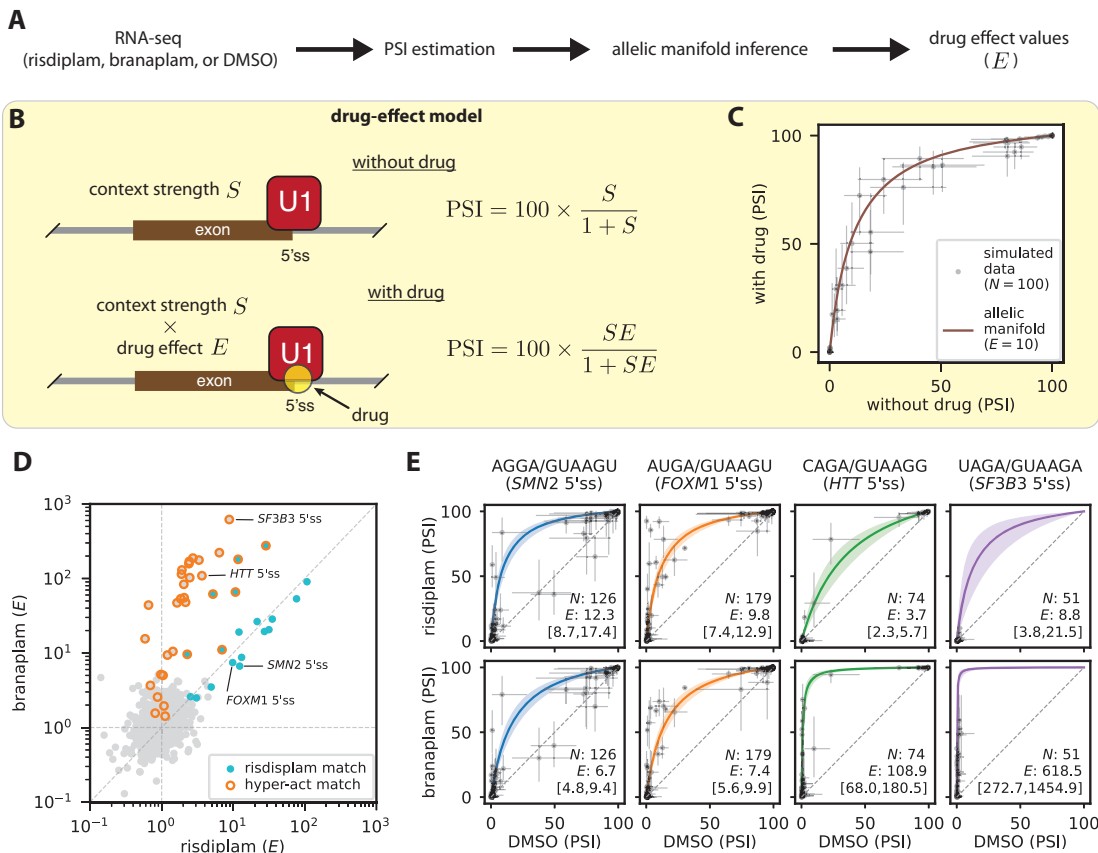

**Fig. 2 | RNA-seq measurements of drug effect are consistent with IUPAC motifs for the 5'ss specificities of risdiplam and branaplam. A** Experimental and computational approach for measuring 5'ss-specific drug effects by RNA-seq. **B** Allelic manifold model for PSI as a function of 5'ss-specific drug effect (quantified by $E$; $E = 1$ in the absence of drug) and locus-specific context strength (quantified by $S$). See Fig. S5 for a derivation of this model as a biophysical model defined by states and Gibbs free energies. **C** Example allelic manifold and simulated RNA-seq data. **D** Scatter plot of drug effects determined for 2521 distinct 5'ss sequences occurring in at least 10 exons identified by rMATS[46]; see SI Sec. 1.7 for details. 5'ss sequences matching the risdiplam IUPAC motif and/or hyper-activation IUPAC motif are indicated. *SMN2* 5'ss: AGGA/GUAAGU, sequence of the 5'ss of *SMN2* exon 7. *FOXM1* 5'ss: AUGA/GUAAGU, sequence of the alternative 5'ss of *FOXM1* exon 9. *HTT* 5'ss: CAGA/GUAAGG, sequence of the 5'ss of *HTT* pseudoexon 50a. *SF3B3* 5'ss: UAGA/GUAAGA, sequence of the 5'ss of *SF3B3* pseudoexon 2a. **E** Allelic manifolds determined for the four 5'ss sequences annotated in panel D. *N*, number of exons identified by rMATS and having the indicated 5'ss. *E*, 5'ss-dependent drug effect inferred by Bayesian curve fitting (median and 95% posterior credible interval). PSI, percent spliced in. 5'ss, 5' splice site. DMSO, dimethyl sulfoxide.

for details. Figure 2D shows the resulting $E$ values determined for risdiplam and for branaplam. Figure 2E shows the inferred allelic manifolds, as well as the underlying PSI values, for the four specific 5'ss sequences indicated in Fig. 2D. Some exons exhibited a statistically significant deviation from their inferred allelic manifold, implying that sequence context outside the 10 bp 5'ss can, at least in some cases, impact drug effect. Nevertheless, visual inspection of Fig. 2E confirms that the inferred allelic manifolds provide good first-order quantitative explanations for the PSIs of genomic exons measured in the presence vs. absence of drug, thereby lending support to our approach for quantifying 5'ss-dependent drug effect.

To validate the $E$ values determined for risdiplam and branaplam, we asked whether these $E$ values were consistent with known drug targets. The 5'ss sequences of four exons known to be affected by risdiplam and/or branaplam are annotated in Fig. 2D; the corresponding allelic manifolds are shown in Fig. 2E. As expected, the $E$ values for risdiplam and branaplam were nearly equal on the 5'ss sequence of *SMN2* exon 7. $E$ values for risdiplam and branaplam were also nearly equal for the alternative 5'ss of *FOXM1* exon 9, an established off-target of risdiplam that is believed to contribute to clinically relevant side effects[15]. Also as expected, branaplam had a much larger $E$ value than risdiplam for the 5'ss of *HTT* pseudoexon 50a, the target against which branaplam has been proposed as a potential treatment

for Huntington's disease[48–51]. And again, as expected, branaplam had a much larger $E$ value than risdiplam for the 5'ss of *SF3B3* pseudoexon 2a, which was recently used to develop a branaplam-inducible gene therapy platform[25]. We conclude that the drug-effect values $E$ determined for risdiplam and branaplam are consistent with the known targets of these two drugs.

### RNA-seq results support a one-motif model for risdiplam and a two-motif model for branaplam

We next investigated whether the $E$ values we determined for risdiplam and branaplam were consistent with the above one-motif model for risdiplam and two-motif model for branaplam. Figure 2D shows the 5'ss sequences matched by the risdiplam IUPAC motif and/or by the hyper-activation IUPAC motif. Consistent with the MPSA data in Fig. 1D, the risdiplam IUPAC motif matches the 5'ss most strongly activated by risdiplam. Consistent with MPSA data in Fig. 1E, the hyper-activation IUPAC motif matches 5'ss activated substantially more by branaplam than by risdiplam. And consistent with the MPSA data in Fig. 1F, the 5'ss most strongly activated by branaplam all match either the risdiplam IUPAC motif or the hyper-activation IUPAC motif. We conclude that both the one-motif model for risdiplam and the two-motif model for branaplam are qualitatively consistent with the drug effects observed across the human transcriptome.

## Biophysical modeling quantitatively characterizes the 5′ss specificities of risdiplam and branaplam

Next we sought to quantitatively characterize the 5′ss specificities of risdiplam and branaplam. To do this, we extended the thermodynamic drug-effect model in Fig. 2B to incorporate three assumptions: (assumption 1) branaplam binds to the U1/5′ss complex in two distinct interaction modes; (assumption 2) one branaplam interaction mode has the same 5′ss sequence specificity as risdiplam; and (assumption 3)

each base in the 10-nt 5′ss has an independent and additive effect on the Gibbs free energy of drug binding to the U1/5′ss complex. Note that additivity assumptions, like assumption 3, are common in thermodynamic models of sequence-dependent interaction energies[41,44,52,53]. The resulting thermodynamic model is illustrated in Fig. 3A. In the absence of drug, pre-mRNA can be in one of two states: not bound by U1 (state 1, Gibbs free energy 0); or bound by U1 (state 2, Gibbs free energy $\Delta G_{U1}$). In the presence of risdiplam, pre-mRNA can be in one of

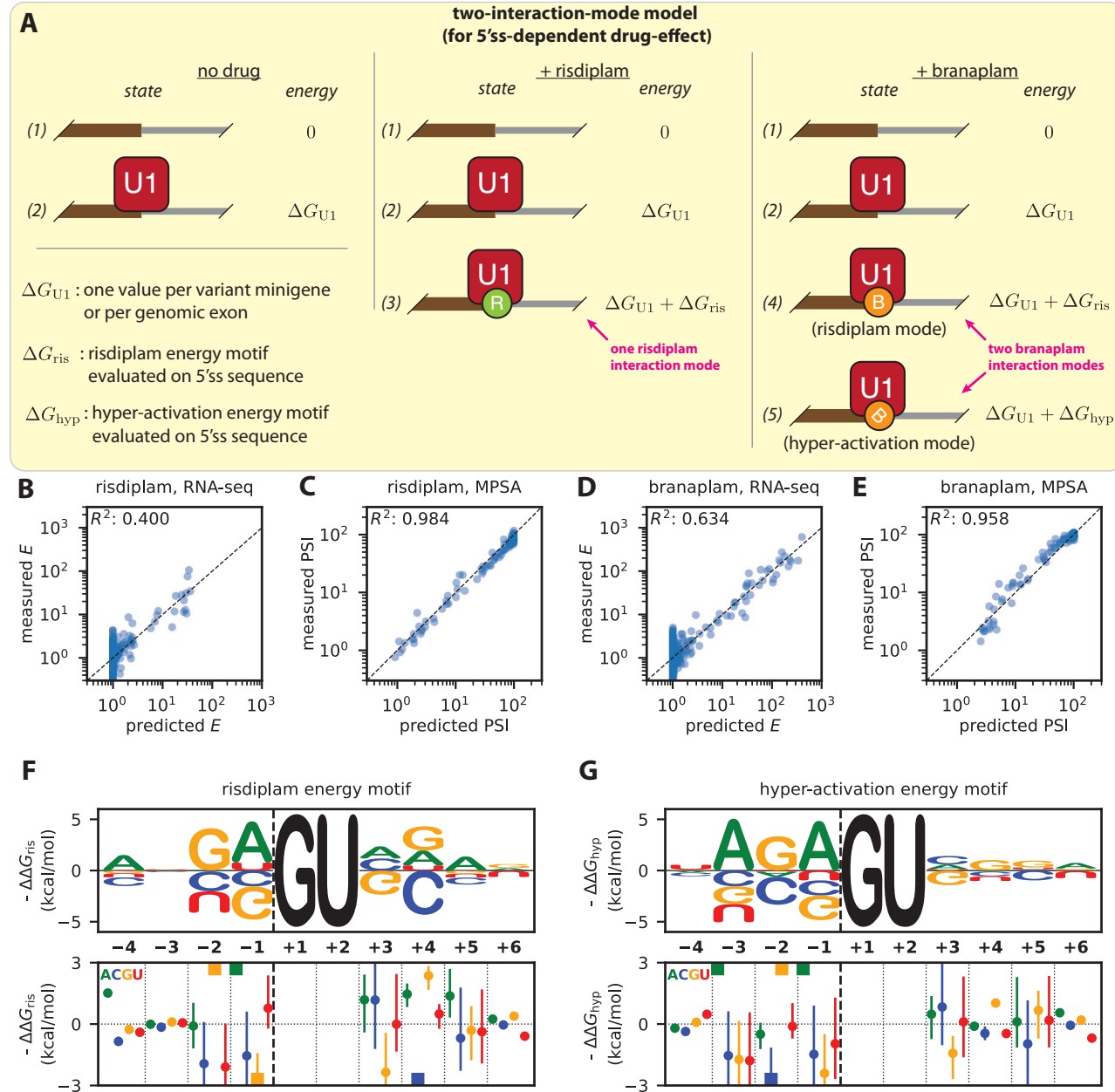

**Fig. 3 | Biophysical model for the sequence specificity of risdiplam and branaplam. A** The "two-interaction-mode model" for how risdiplam and branaplam affect splicing. PSI is assumed to be 100 times the equilibrium occupancy of U1 binding to the 5′ss. Model assumes three sequence-dependent Gibbs free energies: $\Delta G_{U1}$, energy of U1 binding to the 5′ss; $\Delta G_{ris}$, energy of risdiplam binding to the U1/5′ss complex or of branaplam binding to the U1/5′ss complex in the "risdiplam mode"; $\Delta G_{hyp}$, Gibbs free energy of branaplam binding to the U1/5′ss complex in the "hyper-activation mode". Model parameters were inferred from the PSI values measured by MPSA on cells treated with DMSO, risdiplam, or branaplam (Fig. 1D–F), as well as from drug effect values $E$ for risdiplam or branaplam determined by the RNA-seq (Fig. 2D). See text, SI Sec. 3.2, and SI Sec. 4.2 for additional

information. **B–E** Experimentally measured vs. model-predicted PSI values and drug-effect values. PSI values are from the *SMN2* exon 7 MPSA performed on cells treated with risdiplam or branaplam (Fig. 1); drug-effect values are from the RNA-seq analysis in Fig. 2. **F, G** Inferred single-nucleotide effects for (**F**) the "risdiplam energy motif" and (**G**) the "hyper-activation energy motif". Top panels show median parameter values illustrated as sequence logos[71]. Bottom panels show medians (colored dots, with colors corresponding to each of the four RNA bases as indicated) and 95% credible intervals (colored lines) for motif parameters. Colored squares, median values that lie outside the y-axis limits. 5′ss, 5′ splice site. MPSA, massively parallel splicing assay.

three states: state 1; state 2; or bound by U1 and risdiplam (state 3, Gibbs free energy $\Delta G_{U1} + \Delta G_{ris}$). In the presence of branaplam, pre-mRNA can be in one of four states: state 1; state 2; bound by U1 and by branaplam in a "risdiplam mode" (state 4, Gibbs free energy same as state 3); or bound by U1 and by branaplam in a "hyper-activation mode" (state 5, Gibbs free energy $\Delta G_{U1} + \Delta G_{hyp}$). Assumption 1 is reflected in the presence of two branaplam-bound states (state 4 and state 5). Assumption 2 is reflected in state 4 having the same Gibbs free energy as state 3. Assumption 3 is reflected in the assumption that both $\Delta G_{ris}$ (described by a "risdiplam energy motif") and $\Delta G_{hyp}$ (described by a "hyper-activation energy motif") are additive functions of 5'ss sequence; see SI Sec. 3.2 for mathematical details. We refer to this 5'ss-dependent drug-effect model with two branaplam interaction modes as the "two-interaction-mode model".

We then used the MPSA data and RNA-seq data to jointly infer values for the parameters of the two-interaction mode model. Using a Bayesian inference procedure, we sampled posterior values for all 351 parameters of this model, using as data the PSI values measured by MPSA and the drug-effect values $E$ measured by RNA-seq. See SI Sec. 4.2 for details. Plotting the resulting model predictions against the MPSA and RNA-seq data confirms that the two-interaction-mode model explains both MPSA and RNA-seq data well (Fig. 3B–E). We note that higher $R^2$ was obtained on the MPSA data relative to RNA-seq data. This was due, at least in part, to the fact that different sets of 5'ss sequences were assayed by the different methods. The residual deviations between measured $E$ values and predicted $E$ values (Fig. 3B, D) also suggest that risdiplam and branaplam binding energies are not perfectly additive. Nevertheless, we conclude that the two-interaction-mode biophysical model provides a good quantitative description of the 5'ss specificities of risdiplam and branaplam.

Moreover, the energy motifs inferred as part of the two-interaction-mode model largely recapitulated the IUPAC motifs determined in Fig. 1 and Fig. 2. In the exonic region of the 5'ss, the risdiplam energy motif (Fig. 3F) and branaplam energy motif (Fig. 3G) both show a prominent requirement for $G_{-2}$ and $A_{-1}$. The risdiplam energy motif also requires $A_{-4}$ but is agnostic to the base at position −3, whereas the hyper-activation energy motif favors $A_{-3}$ but is largely agnostic to the base at −4. In the intronic region of the 5'ss, the risdiplam energy motif strongly penalizes both $G_{+3}$ and $C_{+4}$—behavior that is again consistent with the risdiplam IUPAC motif (Fig. 1D). We conclude that the two-interaction-mode biophysical model is qualitatively consistent with the observed specificities of risdiplam and branaplam.

To quantitatively test the hypothesis that branaplam binds to U1/5'ss complexes in two distinct interaction modes, we defined a 5'ss-dependent drug-effect model with one branaplam interaction mode (the "one-interaction-mode model") that has the same number of parameters and hyperparameters as the two-interaction-mode model (SI Sec. 3.3 and Sec. 4.3, Fig. S6A). We then applied our Bayesian inference procedure to this model; the resulting model predictions and energy motifs are shown in Fig. S6B–G. The results show that the two-interaction-mode model explains the data much better than the one-interaction-mode model does, corresponding to a log likelihood ratio of 117.4, with a 95% credible interval of [64.4, 172.5] (Fig. S7A). Stratifying by dataset, the results show that the two-interaction-mode model better accounts for the branaplam MPSA data and branaplam RNA-seq data, while similarly accounting for the risdiplam MPSA data, the DMSO MPSA data, and the risdiplam RNA-seq data (Fig. S7B–F). We conclude that quantitative biophysical modeling supports the hypothesis that branaplam interacts with U1/5'ss complexes in two distinct interaction modes, rather than one interaction mode.

## Risdiplam and branaplam exhibit specificity beyond that suggested by the bulge-repair mechanism

NMR structures of the U1 snRNA/5'ss mRNA complex in the presence and absence of SMN-C5 (an analog of risdiplam) led to the suggestion of a "bulge-repair" mechanism for risdiplam activity[20]. In the absence of SMN-C5, the 5'ss $A_{-1}$ bulges out of the minor groove of the U1 snRNA/5'ss double helix (Fig. 4A), destabilizing the helix and potentially clashing with U1-C, a protein component of the U1 snRNP. This $A_{-1}$ bulge is stabilized by the 5'ss $G_{-2}$ pairing in a shifted register with $C_9$ of the U1 snRNA. SMN-C5 binds within the dsRNA major groove, where its carbonyl group forms a hydrogen bond with the amino group of $A_{-1}$, thereby pulling the $A_{-1}$ out of the minor groove and into the helical stack (Fig. 4B). This interaction stabilizes the RNA double helix by ~1 °C, and eliminates the potential clash with U1-C[20]. SMN-C5 is structurally similar to risdiplam (Fig. 4C, D), and is likely to stabilize the U1/5'ss complex by the same mechanism. Branaplam is less similar to SMN-C5 (Fig. 4E), but branaplam does possess a hydroxyl group and a pyridazine group in its center, and either of these groups might be positioned appropriately to form a hydrogen bond with the amino group of $A_{-1}$. Chemical shifts observed in a different NMR study[18] confirm the binding of branaplam to the U1 snRNA/5'ss mRNA complex in the vicinity of $A_{-1}$. These structural studies provide important context for interpreting our biophysical-modeling results.

Our biophysical-modeling results reveal specificity determinants beyond those suggested by the bulge-repair mechanism. The bulge-repair mechanism predicts a critical role for the $G_{-2}pA_{-1}$ dinucleotide. Consistent with this mechanism, both $G_{-2}$ and $A_{-1}$ are strongly required in both the risdiplam energy motif and the hyper-activation energy motif (Fig. 3F,G; see also Fig. S8B,C). However, the bulge-repair mechanism does not obviously explain the requirement of risdiplam activation for an A at position −4 [observed in the risdiplam energy motif (Fig. 3F) and in the raw MPSA data (Fig. S8A)], or the requirement of hyper-activation by branaplam for an A at position −3 [observed in the hyper-activation energy motif (Fig. 3G) and in the raw MPSA data (Fig. S9)]. Indeed, the 5'ss RNA fragment used in the NMR structures of ref. 20 has a G at position −3 and does not extend to position −4. The bulge-repair mechanism also does not explain the preference, in both the risdiplam energy motif and the hyperactivation energy motif, for $G > A > U > C$ at position +4. Low-throughput qPCR measurements (Fig. 4F, G) confirmed that both drugs prefer $G > A > U$, C at position +4 (as expected from Fig. 3F, G; see also Fig. S8E), and further revealed that both drugs prefer $A > C$, G, U at position +3 (see also Fig. S8D) and $G > A$, C, U at position +5. We note that these observations of sequence specificity at positions +3, +4, and +5 are consistent with chemical-shift data for branaplam and two risdiplam analogs reported by ref. 18. We conclude that risdiplam and branaplam exhibit sequence specificity for the $G_{-2}pA_{-1}$ dinucleotide, consistent with the bulge-repair mechanism, but also exhibit additional specificity both upstream and downstream of the $G_{-2}pA_{-1}$ dinucleotide, specificity that is not obviously explained by the bulge-repair mechanism.

## Structural properties of branaplam suggest possible explanations for the two-interaction-mode mechanism

We offer several possible structural explanations for the two putative modes of branaplam-dsRNA interaction at the 5'ss of *SMN2* exon 7. One possible explanation is that branaplam has multiple tautomeric forms (Fig. 4E), and the different tautomeric forms of branaplam might bind the U1/5'ss complex in different orientations. A second possible explanation is that branaplam has two potential hydrogen-bonding groups in its center (Fig. 4E, yellow highlight), and these could allow branaplam to interact with the $A_{-1}$ in two different ways. A third possible explanation is that branaplam has more rotational degrees of freedom than risdiplam (Fig. 4E, green highlight), which could allow branaplam to adopt multiple conformations when interacting with the U1/5'ss complex. Molecular dynamics simulations confirmed that branaplam has greater conformational flexibility than risdiplam does (Fig. S10). A fourth possible explanation is that the approximate dihedral symmetry of dsRNA could facilitate the binding of risdiplam or branaplam in two flipped orientations, and that the sequence

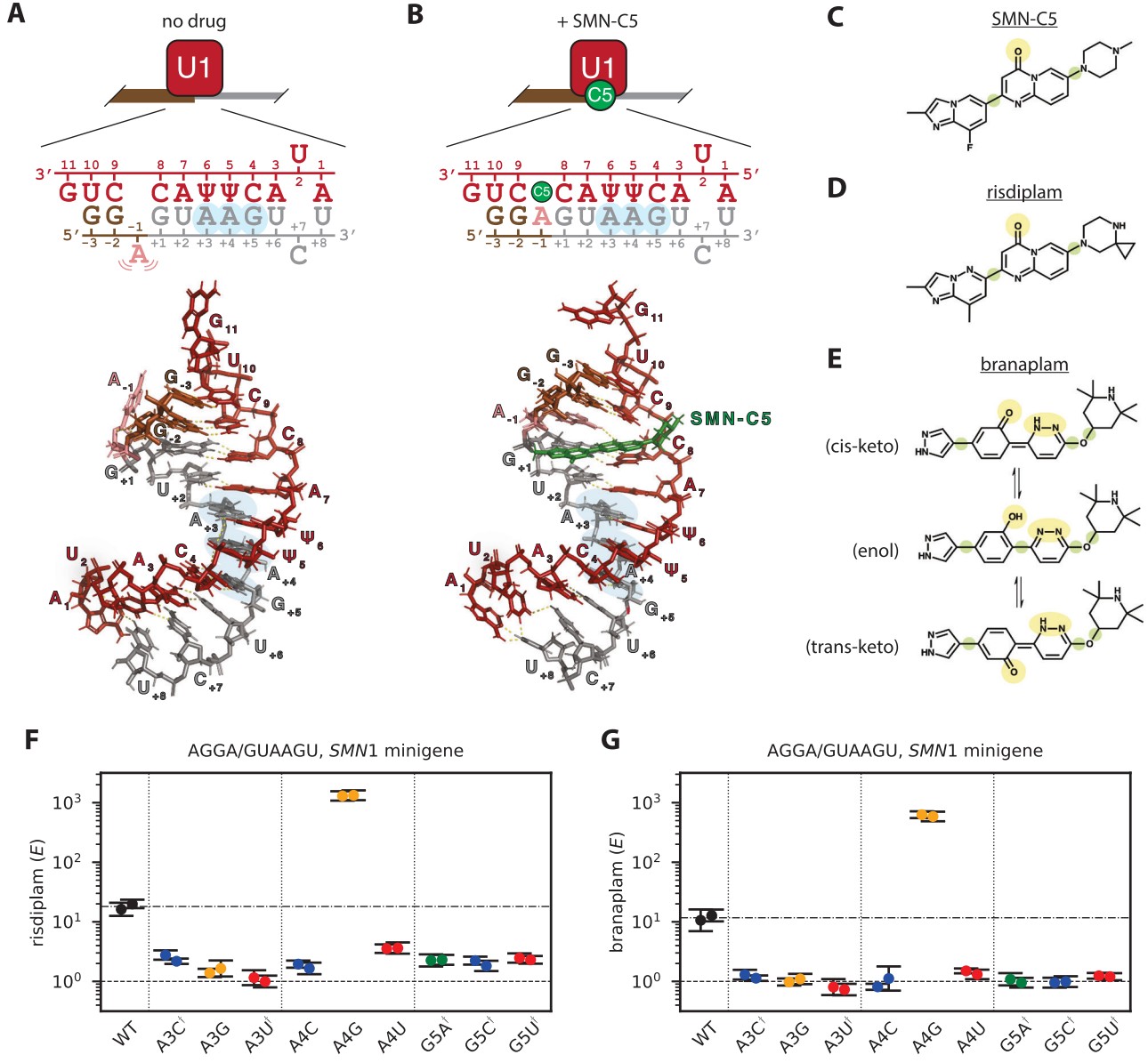

**Fig. 4 | Risdiplam and branaplam specificities are incompletely explained by the bulge-repair mechanism. A, B** Bulge-repair mechanism proposed for the specificity of risdiplam. NMR structures (PDB:6HMI [https://doi.org/10.2210/pdb6HMI/pdb] and PDB:6HMO [https://doi.org/10.2210/pdb6HMO/pdb], from ref. 20) show a U1 RNA/5′ss RNA complex in the (**A**) presence and (**B**) absence of SMN-C5, a risdiplam analog. A schematic of each structure is also shown. Red, U1 snRNA; brown, exonic 5′ss RNA; gray, intronic 5′ss RNA; green, SMN-C5; salmon, bulged $A_{-1}$ stabilized by SMN-C5. Blue highlight, intronic positions observed to affect the activities of risdiplam and branaplam in panels (**F**) and (**G**). **C**–**E** Structures of (**C**) SMN-C5, (**D**) risdiplam, and (**E**) three tautomeric forms of branaplam (cis-

keto, enol, and trans-keto). Yellow highlight, potential hydrogen-bonding partners for the amino group of $A_{-1}$. Green highlight, rotational degree of freedom.
**F, G** qPCR validation of intronic specificities for (**F**) risdiplam and (**G**) branaplam, assayed on the indicated single-nucleotide variants of AGGA/GUAAGU in the context of an *SMN1* minigene. *E* denotes drug effect, which was measured by qPCR as described in SI Sec. 1.8. Note that $E = 1$ corresponds to no drug effect. Dots, $n = 2$ biological replicates; error bars, standard error across $n = 4$ technical replicates; dashed line, no effect; dashed/dotted line, wild-type effect value (geometric mean of biological replicates). Daggers indicate 5′ss variants for which the dominant inclusion isoform uses a cryptic 5′ss at position +52 of *SMN1* intron 7.

specificities of the two flipped orientations of branaplam are more distinct than those of risdiplam. Additional studies are needed to discern which (if any) of these possible explanations are correct.

**Dose-response curves falsify the two-site hypothesis for risdiplam activity at *SMN2* exon 7**

Our two-interaction-mode model for branaplam explains why risdiplam is more specific than branaplam at *SMN2* exon 7. Two publications[18,21] have proposed a different mechanism, which we refer to as the "two-site hypothesis for risdiplam". In addition to one molecule of risdiplam stabilizing the U1/5′ss complex, the two-site hypothesis for risdiplam states that a second molecule of risdiplam

binds a purine tract (PT) within *SMN2* exon 7, thereby synergistically promoting exon inclusion (Fig. 5A). Distinguishing between the two-site hypothesis for risdiplam and the two-interaction-mode hypothesis for branaplam is essential for understanding the molecular mechanisms of risdiplam, branaplam, and splice-modifying small molecules more generally.

A key prediction of the two-site hypothesis is that disrupting the PT should reduce the cooperativity (i.e., Hill coefficient) of the response of *SMN2* exon 7 to risdiplam, but not the cooperativity of the response to branaplam. To test this prediction, we used qPCR to obtain risdiplam dose-response curves for *SMN2* minigenes that carried either the wild-type PT or one of three previously reported PT disruptions:

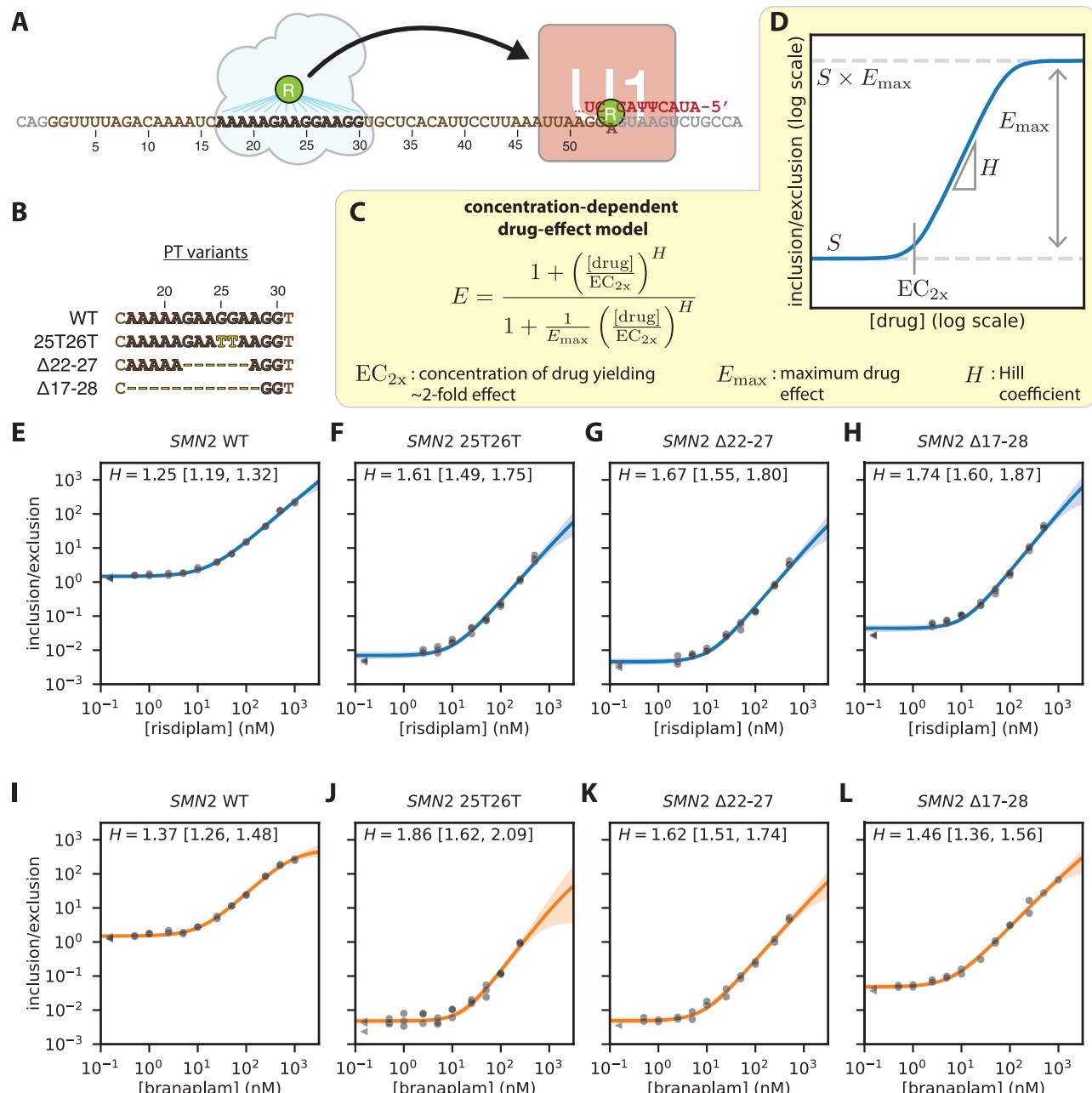

**Fig. 5 | Dose-response curves falsify the two-site hypothesis for risdiplam.**
**A** Two-site hypothesis for risdiplam activity at *SMN2* exon 7. Cloud, proteins hypothesized to mediate the effect of risdiplam at the PT of *SMN2* exon 7 [hnRNP G (ref. 18) or FUBP1 and KHSRP (ref. 21)]. **B** PT variants assayed in *SMN2* minigenes. **C** Empirical model for concentration-dependent drug activity; see also Fig. S11. **D** Schematic illustration of model predictions for the inclusion/exclusion ratio as a function of drug concentration, as well as how model parameters shape this function. **E–H** Risdiplam titration curves for *SMN2* exon 7 minigenes containing (**E**)

the wild-type PT or (**F–H**) mutated PTs. **I–L** Branaplam titration curves for *SMN2* exon 7 minigenes containing (**I**) the wild-type PT or (**J–L**) mutated PTs. Dots and triangles, median qPCR measurements of *n* = 4 technical replicates, shown for *n* = 3 biological replicates at nonzero drug concentration (dots) or zero drug concentration (triangles). Lines and shaded regions, predictions (median and 95% credible interval) of inferred dose-response curves. *H*, inferred Hill coefficients (median and 95% credible interval). PT, purine tract.

25T26T (ref. 54), Δ22–27 (ref. 21), and Δ17–28 (ref. 18); see Fig. 5B. We then used these data to infer the parameters of an empirical quantitative model for concentration-dependent drug effect (Fig. 5C, D), which builds on the drug-effect model of Fig. 2B. This model has four parameters: (i) context strength of the target exon ($S$); (ii) maximal drug effect ($E_{max}$); (iii) concentration of two-fold drug effect ($EC_{2x}$); and (iv) Hill coefficient ($H$). Note that this model is parameterized using $EC_{2x}$ instead of the more standard $EC_{50}$ (the concentration of drug that yields PSI halfway between basal PSI and saturation PSI); the reasons for this are presented in SI Sec. 3.4. After defining this model, a

Bayesian inference procedure was used to infer values for these four parameters from each dose-response curve (SI Sec. 4.4). The resulting parameter values are shown in Table S1. Contrary to expectations under the two-site hypothesis, mutating the PT increased (rather than decreased) the Hill coefficient of risdiplam at *SMN2* exon 7 (Fig. 5E–H). A similar effect was observed for branaplam (Fig. 5I–L). Measurements for a non-target exon (*ELP1* exon 20) confirmed that the observed dose-response behavior was specific to *SMN2* exon 7, and thus did not reflect global changes in spliceosome behavior (Fig. S12A, B). Moreover, the risdiplam and branaplam Hill coefficients were quantitatively

similar, regardless of whether or not the PT is present. We conclude that the dose-response data contradict the cooperative formulation of the two-site hypothesis.

The empirical modeling results in Fig. 5 provide an alternative explanation for data, reported in two prior studies[18,21], that were previously interpreted as supporting the two-site hypothesis for risdiplam. Both prior studies measured dose-response curves for risdiplam analogs (SMN-C3 and SMN-C5, respectively) in the context of *SMN2* minigenes having the wild-type PT or deletions in the PT (Δ17–28 and Δ22–27, respectively). Both prior studies observed that PT deletions increased $EC_{50}$, and both prior studies interpreted this result as evidence for reduced cooperativity in the response of exon 7 to risdiplam analogs. Our quantitative model, however, shows that $EC_{50}$ is affected by multiple parameters (SI Sec. 3.4). Genetic changes that affect $EC_{50}$ can therefore be caused by changes in any of these four parameters, and therefore reflect different molecular mechanisms. When applied to our dose-response data, our quantitative model reveals that PT mutations strongly reduce context strength (quantified by $S$) while modestly increasing, not decreasing, cooperativity (quantified by $H$). This reduction in context strength fully accounts for the increases in $EC_{50}$ that both prior studies observed when deleting the PT, and is consistent with the fact that the splicing activator Tra2-β1 is known[55] to bind the GGA sequence at positions 25–27 of exon 7. A similar critique applies to a more recent study[31], which also assayed the effects of risdiplam and branaplam in the presence of mutations in the interior of *SMN2* exon 7. We propose that similar reductions in context strength likely account for the increase in $EC_{50}$ observed by one of these prior studies[21] upon knocking down FUBP1 and KHSRP, two proteins that were proposed to bind the PT and mediate risdiplam-dependent splicing activation. We conclude that the prior observed effects of PT disruptions in cells are explained by the PT acting as a risdiplam-independent, rather than risdiplam-dependent, splicing enhancer. We also conclude that empirical quantitative models can play a valuable role in mechanistic studies of splice-modifying drugs.

### Anomalous cooperativity is a common feature of splice-modifying drugs

Our dose-response curves unexpectedly revealed that risdiplam and branaplam exhibit substantial cooperativity, i.e., have Hill coefficients greater than one. We refer to this phenomenon as "anomalous cooperativity" because it is unlikely to result from the simultaneous binding of multiple drug molecules. Anomalous cooperativity is interesting, both because of the mechanistic questions it raises and because it has the potential to impact drug specificity. To understand if anomalous cooperativity occurs more generally, we investigated the dose-response behavior of other splice-modifying drugs.

We first asked if other drugs that promote the inclusion of *SMN2* exon 7 exhibit anomalous cooperativity. We measured dose-response curves for two antisense oligonucleotides, ASOi7 and ASOi6[11]. ASOi7 binds to *SMN2* intron 7 at positions +9 to +23 (relative to the 5′ss of exon 7), and presumably functions by the same mechanism as nusinersen, i.e., by blocking RNA binding by the splicing repressor hnRNP A1/A2. ASOi6 binds *SMN2* intron 6 at positions −55 to −41 (relative to the 3′ss of exon 7), and may function by blocking RNA binding by the splicing repressor HuR[56]. The results show that ASOi7 exhibited anomalous cooperativity, similar to risdiplam and branaplam (Fig. 6A). The results also show that ASOi6 exhibited substantially less (if any) cooperativity (Fig. 6B). Measurements for a non-target exon (*ELP1* exon 20) confirmed that the observed dose-response behaviors of ASOi7 and ASOi6 were specific to *SMN2* exon 7 (Fig. S12C, D). A BLASTN[57] analysis confirmed that ASOi7 does not have significant complementarity to off-target sites within the *SMN2* minigene pre-mRNA. ASOi6 does have partial complementarity to nearby sites, but has substantially lower Hill coefficient than ASOi7. We conclude that a splice-modifying drug can exhibit cooperativity even when it binds

only a single site on target pre-mRNA, and that the extent of this cooperativity can differ between drugs that promote inclusion of the same cassette exon.

We then asked if splice-modifying drugs that promote the inclusion of exons other than *SMN2* exon 7 also exhibit anomalous cooperativity. We therefore measured dose-response curves for two drugs, RECTAS[58] and ASOi20[59], previously developed as potential treatments for familial dysautonomia. Both RECTAS and ASOi20 promote the inclusion of *ELP1* exon 20. RECTAS is a small molecule that was proposed to function by indirectly enhancing the phosphorylation of the splicing factor SRSF6[60]. ASOi20 binds to *ELP1* intron 20 at positions +6 to +20 (relative to the 5′ss of exon 20), and functions through an unknown mechanism, presumably involving blocking of an intronic splicing silencer[59]. Our results show that RECTAS exhibited significant cooperativity (Fig. 6C), whereas ASOi20 exhibited less (if any) cooperativity (Fig. 6D). Measurements for a non-target exon (*SMN2* exon 7) confirmed that the observed dose-response behavior was specific to *ELP1* exon 20 (Fig. S12E, F). We conclude that anomalous cooperativity is a common feature of many (but possibly not all) splice-modifying drugs.

### Multi-drug synergy is another common feature of splice-modifying drugs

We next asked whether two distinct drugs that promote inclusion of the same cassette exon, but which bind to distinct molecular targets, could synergistically promote exon inclusion. There are many different mathematical definitions for synergy[61], but the operative definition for two-drug cocktails is whether a mixture of two drugs can yield a greater effect than either drug alone. To test for synergy, we measured exon inclusion/exclusion ratios in response to different linear mixtures of drugs, i.e., mixtures of two drugs that interpolate between two single-drug endpoints: (100% drug 1, 0% drug 2) and (0% drug 1, 100% drug 2). Under the no-synergy hypothesis, one should observe a monotonic response that interpolates between the two single-drug endpoints. On the other hand, if the inclusion/exclusion ratio is maximal at an intermediate mixture, this would indicate the presence of synergy between the two drugs. We therefore tested for synergy based on whether any intermediate drug mixture yielded an inclusion/exclusion ratio higher than both of the single-drug endpoints. P-values were computed using an empirical (i.e., non-mechanistic) Bayesian model described in SI Sec. 4.5. The results show no significant synergy between risdiplam and branaplam at *SMN2* exon 7 (Fig. 6E). This observation is consistent with risdiplam and branaplam binding the same or overlapping sites in the *SMN2* exon 7 U1/5′ss complex. By contrast, the results show significant synergy among all other pairs of drugs that target *SMN2* exon 7 (Fig. 6F–J). The results also show strong synergy at *ELP1* exon 20 between RECTAS and ASOi20 (Fig. 6K). We confirmed these findings using a separate test for synergy, i.e., one based on the Hill coefficients of dose-response curves measured for two-drug cocktails (Fig. S13 and Table S1). We conclude that synergy is widespread (and possibly universal) among splice-modifying drugs that can simultaneously promote inclusion of the same cassette exon.

### Discussion

We have introduced and applied quantitative models for splice-modifying drugs. We used these quantitative models to analyze data from MPSAs, RNA-seq experiments, and precision dose-response curves. The results quantitatively characterize the 5′ss specificities of risdiplam and branaplam. The results also change the mechanistic understanding of how these two small-molecule drugs function. Contrary to the prevailing two-site hypothesis for risdiplam[18,21], our results show that risdiplam does not promote *SMN2* exon 7 inclusion, even in part, by binding to a purine tract within *SMN2* exon 7. The results also suggest that branaplam binds U1/5′ss complexes in two distinct interaction modes: one interaction mode that confers specificity

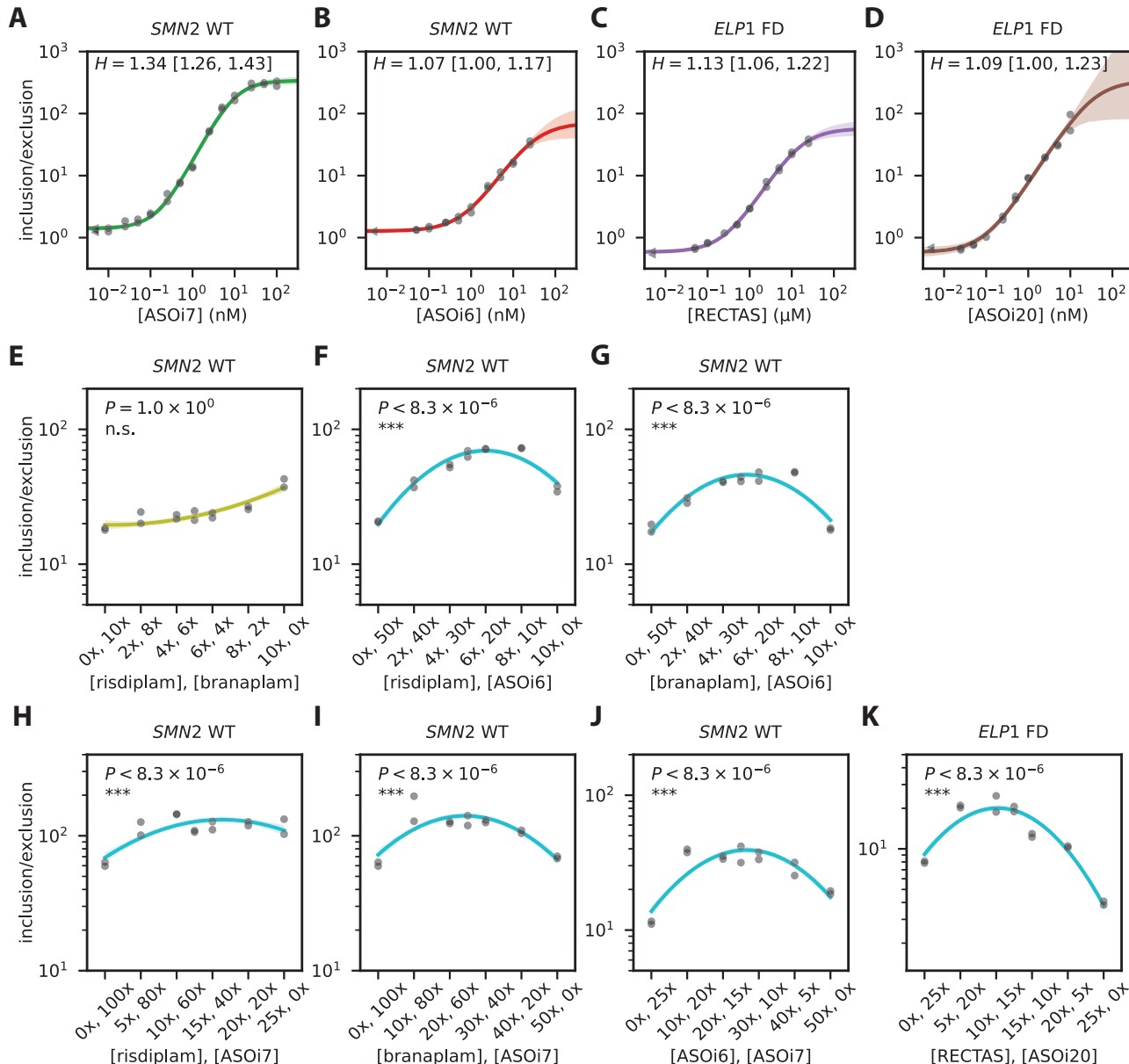

**Fig. 6 | Anomalous cooperativity and multi-drug synergy among splice-modifying drugs. A, B** Single-drug dose-response curves for *SMN2* exon 7 in response to (A) ASOi7 and (B) ASOi6. **C, D** Single-drug dose-response curves for *ELP1* exon 20 in response to (**C**) RECTAS and (**D**) ASOi20. Dots, median qPCR measurements of n = 4 technical replicates, shown for n = 2 biological replicates. Lines and shaded regions, predictions (median and 95% credible interval) of inferred dose-response curves. **E–J** Two-drug linear-mixture curves measured for *SMN2* exon 7 in response to (**E**) risdiplam/branaplam mixtures, (**F**) risdiplam/ASOi6 mixtures, (**G**) branaplam/ASOi6 mixtures, (**H**) risdiplam/ASOi7 mixtures, (**I**) branaplam/ASOi7 mixtures, and (**J**) ASOi6/ASOi7 mixtures. **K** Two-drug linear-mixture curves measured for *ELP1* exon 20 in response to RECTAS/ASOi7 mixtures. In panels E-K, curves are second-order polynomials fit to the data points shown using a Bayesian inference procedure (described in SI Sec. 4.5). 1x concentration of each drug (corresponding to approximate $EC_{2x}$ values) is 14 nM for risdiplam, 7 nM for branaplam, 0.6 nM for ASOi6, 0.1 nM for ASOi7, 300 nM for RECTAS, and 0.08 nM for ASOi20. *P*, p-value for no-synergy null hypothesis (i.e., that the maximal inclusion/exclusion ratio occurs at one of the two ends of the mixture curve) computed using Hamiltonian Monte Carlo sampling as described in SI Sec. 4.5. \*\*\*, *P*<0.001; \*\*, *P*<0.01; \*, *P*<0.05; n.s., *P* ≥ 0.05.

similar to risdiplam's (the risdiplam mode), and a second interaction mode that confers alternative sequence specificity (the hyperactivation mode). The results further show—remarkably—that single-drug cooperativity and multi-drug synergy are widespread in the dose-response behavior of splice-modifying drugs. These findings establish quantitative modeling as a powerful tool in the study of splice-modifying drugs, reveal details about the mechanisms of two clinically important drugs, and suggest new approaches for developing therapeutics.

Our biophysical modeling approach provides a new way to study the specificities of splice-modifying drugs. Prior studies have reported

position weight matrices (PWMs) for genomic 5'ss sequences that are activated by risdiplam or branaplam[17,25,50]. However, these PWMs convolve two fundamentally different signals: the sequence features that a 5'ss must have to be activated by a drug, and the sequence features a 5'ss must have to be functional in the absence of the drug (see Fig. S14). Our approach deconvolves these signals, and is thus able to quantify the energetic effects of individual 5'ss nucleotides on the binding of risdiplam or branaplam to the U1/5'ss complex.

Our biophysical modeling results provide mechanistic insight into how risdiplam and branaplam interact with the U1/5'ss complex. The 5'ss specificities of risdiplam and branaplam are consistent with

a bulge-repair mechanism[20], in that both drugs require a non-canonical $G_{-2}pA_{-1}$ dinucleotide in the target 5'ss. However, both risdiplam and branaplam exhibit additional specificity upstream and downstream of the $G_{-2}pA_{-1}$ dinucleotide, specificity that is not obviously explained by existing structural data. For example, we find that the hyper-activation energy motif strongly favors an A at position −3 (which might base-pair with $U_{10}$ of the U1 snRNA), and that the risdiplam energy motif favors an A or G at position +3 (which might base-pair with $\Psi_6$ of the U1 snRNA). Our results also suggest specific RNA sequences that, if used in structural studies similar to that of ref. 20, could provide direct evidence for or against the two-interaction-mode model for branaplam. In particular, we predict that branaplam should bind in a different way to the U1/5'ss complex when the 5'ss sequence contains an $A_{-4}pB_{-3}$ dinucleotide as opposed to a $B_{-4}pA_{-3}$ dinucleotide (where B denotes any nucleotide other than A). More generally, our finding that two independently developed small-molecule therapeutics target the same non-canonical U1/5'ss complex, and that doing so confers an unexpectedly high level of specificity, underscores the importance of non-canonical 5'ss (e.g., 5'ss containing shifted registers or asymmetric bulges[62,63]) as targets for future therapeutic development.

Our quantitative empirical modeling of the dose-response behavior of individual splice-modifying drugs revealed that such drugs often exhibit cooperativity, i.e., have Hill coefficients greater than one. We describe this cooperativity as "anomalous", because it appears not to be due to multiple drug molecules simultaneously binding to the same RNA/protein complex. One possible explanation for the widespread anomalous cooperativity of splice-modifying drugs is the presence of biochemical feedback that couples the splicing of multiple pre-mRNA transcripts from the same gene (e.g., through effects on chromatin[64]). A second possible explanation for this widespread anomalous cooperativity is kinetic proofreading during the splicing of individual pre-mRNA transcripts[65–67], as kinetic proofreading could cause the binding of a drug to its target to be effectively "read out" by the spliceosome more than once. Indeed, multiple DExD/H-box ATPase components of the spliceosome are known to play important roles in ensuring spliceosome fidelity[65]. Although kinetic models of gene regulation can yield Hill coefficients larger than those of analogous thermodynamic models[68], specific mathematical models of kinetic proofreading in splicing would need to be analyzed to determine if such models can account for the anomalous cooperativity we observe. More generally, additional quantitative studies will be needed to establish the roles, if any, that feedback and proofreading play in sensitizing pre-mRNA transcripts to the effects of splice-modifying drugs.

Our quantitative empirical modeling of the dose-response behavior of mixtures of splice-modifying drugs further revealed widespread synergy between drugs that can simultaneously promote inclusion of the same cassette exon. We expect that other alternative splicing processes, such as alternative 5'ss usage, alternative 3'ss usage, intron retention, and mutually exclusive cassette exons, might be affected by splice-modifying drugs in similar synergistic ways. Developing quantitative models that can predict (rather than just describe) this synergistic behavior is an important goal for future work. Such models have the potential to provide mechanistic insight into the stages of spliceosome assembly and mRNA processing that splice-modifying drugs target. Such models also have the potential to guide the development of drug-cocktail therapies.

Our findings have immediate implications for the understanding and development of new splice-modifying therapies. First, our experimental and analytical approach can be used to quantitatively characterize the specificities of other splice-modifying small molecules. Obvious candidates include kinetin[69], RECTAS[58], and BPN-15477[70], which have been developed as potential treatments for familial dysautonomia. Second, our two-interaction-mode biophysical model for branaplam suggests that the specificity of branaplam for pseudoexon 50a in *HTT*[48,50,51] might be increased, and thus clinical side-effects decreased, by chemical changes that stabilize the hyper-activation interaction mode or eliminate the risdiplam-like interaction mode. Third, our observation of widespread multi-drug synergy should motivate the clinical investigation of splice-modifying drug cocktails. We note that drug effects need to be large for this type of synergy to have a substantial effect on PSI. Drug cocktails that leverage this type of synergy are therefore likely to be most therapeutically useful for promoting mRNA isoforms that naturally occur at very low levels (e.g., inclusion of pseudoexon 50a in *HTT*).

More generally, our work establishes the value of quantitative modeling in the study of splice-modifying drugs. Quantitative models based on statistical mechanics and chemical kinetics have, over the last century, played a central role in enzymology and in the development of protein-targeted therapeutics[32–34]. The complexity of the spliceosome, however, has prevented the direct application of these models to splice-modifying drugs. Here we showed that relatively simple quantitative models (both biophysical models and empirical models) can, nevertheless, be useful for studying splice-modifying drugs, including for illuminating the molecular mechanisms of drug action. In addition to facilitating the development of new therapies, similar quantitative modeling strategies will likely facilitate the use of splice-modifying drugs in quantitative studies of the fundamental mechanisms of alternative mRNA splicing.

## Methods

Here we summarize the materials and methods used in this work; details are provided in SI. SI Sec. 1 describes experimental methods. All experiments were performed in HeLa cells, which were cultured as described in SI Sec. 1.1. RNA extraction was performed as described in SI Sec. 1.2. Individual *SMN2* minigene plasmids were constructed as described in SI Sec. 1.3. *SMN2* minigene libraries were constructed as described in SI Sec. 1.4. MPSA experiments were performed on the *SMN2* minigene libraries constructed in this work, as well as on *ELP1* minigene libraries constructed in ref. 36, as described in SI Sec. 1.5. Radioactive gels were performed as described in SI Sec. 1.6. RNA-seq experiments were performed as described in SI Sec. 1.7. In particular, RNA-seq data were analyzed using rMATS[46] and custom Python scripts. qPCR assays were performed as described in SI Sec. 1.8. Dose-response and linear-mixture experiments were performed as described in SI Sec. 1.9. SI Sec. 2 provides details about MPSA data processing, the exploratory IUPAC motif analysis, and molecular dynamics simulations. SI Sec. 3 provides formal mathematical definitions of the quantitative models used in this work. SI Sec. 4 describes the Bayesian inference procedures used to infer the parameters of these quantitative models. Bayesian model inference was carried out in STAN and in Python using numpyro. Plasmids, primers, and key resources are listed in Tables S2–S4. All unique biological materials are available upon request.

### Reporting summary
Further information on research design is available in the Nature Portfolio Reporting Summary linked to this article.

## Data availability
The MPSA and RNA-seq data have been deposited in the NCBI GEO database under accession code GSE221868.

## Code availability
Data-analysis scripts have been deposited on GitHub at https://github.com/jbkinney/22_drugs. A snapshot of this repository is available on Zenodo at https://doi.org/10.5281/zenodo.8353692.

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

## Acknowledgements

We thank Christopher Trotta (PTC Therapeutics), Leemor Joshua-Tor (CSHL), and John Moses (CSHL) for helpful conversations. This work was supported by a JSPS Overseas Research Fellowship (Y.I.), an Interdisciplinary Scholar in Experimental and Quantitative Biology fellowship from the Simons Center for Quantitative Biology at CSHL (M.S.W.), the Simons Foundation (S.M.H.), NIH grant R35 GM133613 (C.M.-G., D.M.M.), the Alfred P. Sloan foundation (D.M.M.), NIH grant R37 GM42699 (Y.I., M.S.W., A.R.K.), NIH grant R35 GM133777 (M.S.W., A.A., J.B.K.), NIH grant R01 HG011787 (M.K., J.B.K.), and additional funding from the Simons Center for Quantitative Biology at CSHL (D.M.M., J.B.K.). Shared resources at CSHL were supported by NIH grants P30 CA045508, S10 OD020122, and S10 OD28632.

## Author contributions

M.S.W., Y.I., C.M.-G., A.A., A.R.K., and J.B.K. designed the research. Y.I., M.S.W., and A.A. performed the experiments. M.K., C.M.-G., and J.B.K. analyzed the data. S.M.H. performed the molecular dynamics simulations. J.B.K., Y.I., C.M.-G., and S.M.H. wrote the manuscript with additional contributions from M.S.W., A.A., M.K., D.M.M., and A.R.K.

## Competing interests

A.R.K. is an inventor on issued nusinersen patents licensed by CSHL to Ionis Pharmaceuticals and Biogen; an inventor on an issued patent for familial dysautonomia ASOs; a co-founder, Director, Chair of the SAB, and shareholder of Stoke Pharmaceuticals; a paid consultant for Biogen; a collaborator of Ionis Pharmaceuticals; and a member of the SABs and shareholder of Skyhawk Pharmaceuticals, Envisagenics, Autoimmunity BioSolutions, and assembl.cns. Other authors declare no competing interests.
