## [Peer Review File · Nature Communications]

Specificity, synergy, and mechanisms of splice-modifying drugsREVIEWER COMMENTS

Reviewer #1 (Remarks to the Author):

The authors of this paper present an interesting approach based on RNA sequencing and data modeling to understand the sequence basis for the mechanism of action of two small molecule drugs, Risdiplam and Branaplam. This is an intensive field of research considering the few small molecules that act on RNA. Several modes of action have been proposed based on structural work and one mutagenesis but a complete understanding of the mode of action and in particular why so few alternative exons are targeted by these molecules that deeper understanding is needed.

Here the authors have elucidated the sequence motifs that are important for the enhancing splicing activity of the two molecules. They also discovered two sequence motifs for branaplam. This sequence motif analysis confirms the importance of a bulge adenine in position -1 but also other key residues in the 5' splice-site sequence. This suggests that they're more than a -1 A bulge. Moreover, they could show evidence for a synergy between several molecules, cooperativity and also that the purine-rich region upstream of the 5' splice site may not be a binding site for these small molecules.

Overall, this is a very important study that gives some clues on the sequence-specific needs for the small molecules to act.

I do not have any major criticisms. Maybe the sequence motifs found could be discussed a little bit more. For example, the -2G maybe important to allow the formation of the bulge A. The -3A can base-pair with the opposite U of U1 snRNA and this might explain why this is needed. This would mean that branaplam requires a more stable helix. Same for the preference for A/G in position +3, this would favor pairing for a pseudo U of U1 snRNA. I think this is worth commenting on in the discussion.

Reviewer #2 (Remarks to the Author):

In this paper, Ishigami et al. develop a quasi-biophysical model to quantitatively describe the influence of splice-modulating drugs (risdiplam, branaplam, and ASOs) on splicing. They lay out models that relate U1 snRNP and drug occupancy to observed splicing outcomes (PSI) in an effort to quantitatively describe the drug specificity. They use these models to propose, interpret, and evaluate models for drug binding and action. This level of quantification of drug effects is generally missing from the field and reflects a shift back to biochemical and biophysical fundamentals in the modern era of cellular studies and high-throughput sequencing that will be necessary to move the field forward.

Overall the approach is an important—even critical—one, which others will likely follow in studying splicing and other aspects of RNA metabolism. Nevertheless, there are major limitations in presentation, data, analysis and interpretation:

The models are not clearly presented. As I understand it the authors are creating a logo or position weight matrix in a two-step process—what sequences determine splice site choice and then what sequences determine whether drug binding will alter the splice site choice. They then create a binding model based on an assumed independence of the terms in the sequence logo, trying multiple versions of defining those logos. If the above description is accurate, the authors should clearly lay this out, along with the assumptions. If it is not accurate, a clearer description is needed, at least for this reviewer. Note that the assignment of logo sequences seems qualitative and thus to an extent arbitrary. This is a significant limitation that minimally needs to be clearly laid out.

In addition, the models presented need to show the splicing event and how that is affected by U1 and drug binding. I think there are unstated assumptions in the model (beyond site independence) and the authors should lay out a full molecular model and then derive an expression from the model. I.e., directly connect the math to the model that is presented.

The analysis allows a non-unit Hill slope, but the presented biophysical models do not. If the Hill slope is empirical, then the model becomes empirical, not biophysical. Further, showing that a biophysical model does not hold is valuable and then proposing (as was done later in the manuscript) why a

model may not hold is also valuable. There more can be done to actually make the models and show what is needed from the revised models to account for the data.

The data themselves are not presented with clarity and thoroughness needed to evaluate to what extent limitations in the model vs. in the data are responsible for deviations, and it is unclear at least to me what errors or deviations should be taken as invalidating the model vs. being within experimental error (either random or systematic). (I.e., what error models are used, in detail?) In particular the absence of saturation in most drug 'binding' curves need to be made clear by presenting sufficient underlying data. Also, actual PSI values rather than normalized (as described in prior papers referenced for the method) should be given and the reader should be able to connect these values to the number of sequence reads and error estimates from all sources.

I appreciate the efforts to lay out quantitative models laid out here, yet the conclusions drawn from those models need further tests before publication and the manuscript as a whole requires further clarification. In particular the models are not clearly laid out, include features and empirical assumptions beyond the biophysical, and the presentation is unclear in several places.

In addition to the above, I suggest the following major and minor points that I believe will strengthen the paper overall.

Major Points:

Generally, the authors claim to have attained quantitative definition of sequence-dependent splicing, understanding of the mechanism of drug action, and novel strategies to apply therapeutically. It is not clear to me that these claims are met. What is the definition of "quantitatively defined" and what is the metric? The data appear to give Hill slopes $\neq 1$ but the mechanisms in the presented biophysical models do not predict that.

MPSA and motif description:

Line 87-88: The goal of finding the sequence features of a functional 5'ss and separating that from the sequence features necessary for drug binding is central to this study. This point could be made clearer by including the logo motif from the DMSO control and then stating what bases change in the motif for drug binding.

The criteria for classifying the 5'ss that are activated by drug, insensitive, or undeterminable seem arbitrary as presented in the methods. How were the regions defined? What happens as those criteria are shifted? To determine the hyp motifs, a ratio of PSI_{branaplam}/PSI_{risdiplam} was used. I suggest considering including a plot of the risdiplam/branaplam ratio.

In Figure 1D-I, there are several sequences that match the motif but are in a region with little to no splicing effect in the DMSO control or with drug treatment (gray cloud). I suggest that the authors elaborate on this as it could also be related to their point about a functional 5'ss and be used to further refine the model.

I'm not sure I follow the logic of defining min and max motifs. It seems from the MPSA that the min motifs are sufficient to describe any data points classified as having an effect. Moreover, the authors derived the motifs with different algorithms. I think one can consign max motifs to supplemental information and reduce the discussion thereof, focusing on the min motifs as starting points for the subsequent experiments. Most fundamentally, defining these motifs is arbitrary and provides the authors multiple 'shots on goal' to come up with a model that fits. Stated another way, a model based on a motif is not really biophysical unless the motif is used to define affinities; it seems that this is (sort of—see below) used to quantitatively predict affinities and then splicing outcome but one needs to dig through the model details to figure this out (see point 2 below).

As presented, it is highly unclear how PSI is determined. The authors must clarify exactly what they did in the Methods; merely referring to another publication is insufficient.

Presentation (see also below): Motifs are used to make predictions but the logic and workflow is not defined and the reader needs to read the Methods (supplement) to figure this out. That said, there also appear to be arbitrary cut-offs used to define different levels of motifs—that are later used to see which gives better fits? If this is not the case, the authors should clarify. If it is, they should present metrics for the agreement of the data and model using each cut-off or some better way to define the

logo, and thus residue weighting, in the model.

Coarse-grained biophysical model

The biophysical models are partial. States and weights are defined, but the partitioning between splice products is not shown in the models. I think I figured this out, but it should be shown.

Lines 263-265: It is not an assumption of the model that each state has a weight, but just how Boltzmann statistics work to describe any system at thermodynamic equilibrium, which is what seems to actually be assumed. I suggest revising to clarify.

Lines 268-269: What the authors seem to be saying is that the energetic contributions ($= -k_B T \ln W_x$, where W_x is a Boltzmann weight) of a given 5'ss to drug/U1 binding are the sum of the energetic contributions of the constituent nucleotides. I think this is a clearer and more interpretable way of stating things as opposed to "the logarithms of the weights... are taken to be additive functions of the 5'ss sequence". One may also point out that energetic additivity is a common, if implicit, underlying assumption of sequence motif descriptions of macromolecule binding. See Jarmoskaite et al. (2019), Mol. Cell for similar statements.

I suggest clarifying the models by combining them where applicable. In Figure 2A, for example, there is actually only one model, but the case without the drug has $E = 1$ (i.e., as all weights $W_x = 0$ identically since $[\text{drug}] = 0$). In general, I think it would clarify the text and improve interpretations to be more explicit about what these weights are in terms of free energies and drug concentrations, which are easier to understand/interpret, and more accessible to a general biology/biochemistry audience.

In this framework, S should be a function of the free concentration of U1 snRNP and its affinity for the site. Assuming the former stays the same, do fit values for S match what we know about U1 binding preferences? Do they teach us anything about these?

Why are branaplam and risdiplam given the same weight for the risdiplam interaction mode when they are used at different concentrations and may have different affinities for the same interaction site? What happens if this assumption is shifted?

Figure 3B: is this for all of the data or bran/ris specific?

Figure 3C-F: Why is R^2 for E so much lower than PSI ? It seems like points with low predicted E values have higher variance. I suggest that the authors comment on this.

Figure S12A-D has almost identical R^2 values for branaplam: What other evidence suggests that this model is correct?

Figure 3G,H: what are the differences between this model and the logo motifs found in Figure 1? I suggest that the authors further elaborate on this.

How many total parameters does each model have? Are they truly additive or does this assumption need testing? How do train/test errors compare to the error in the entire dataset?

Hill coefficient, cooperativity, and titration.

There is no term in the model(s) that predicts a Hill coefficient $\neq 1$, so that using other values turns a model into an empirical fit. The authors do later suggest that there may be proofreading or other factors that give different behavior but do not derive mathematical formulas from such models to show whether (or not) such a model can account for the data and what other properties it would have. I don't fully understand the logic of setting the weight of the drug bound to the RNA alone, A , equal to B/E_{max} . I ask that the authors please clarify and explain in the SI (this also relates to the above main point about more clearly defining biophysical/molecular models that connect to the math).

The model predicts that as one titrates the drug, the drug effect E will increase linearly with the concentration of drug until all splice sites are saturated, after which it will not change. Is this the case? Are some sites predicted to be already saturated based on E values?

FIGURE 5: How is E_{max} fit? Is there a specific reason why higher $[\text{drug}]$'s were not included (e.g., toxicity)? (i.e., the data do not appear to saturate in most cases as would be needed to specify E_{max} . The main data (Figures 5 & 6) are problematic in that an endpoint is not well defined for nearly all the data. While fits often give seemingly good fits, without obtaining a clear plateau, these values can be off by orders of magnitude—and it is not clear how such errors would affect the conclusions.

Isn't the fact that disrupting the PT increased the Hill coefficient evidence that it also binds there and just doesn't have the intended effect on splicing? E.g., the effective concentration is higher at the functional site. Another model is that binding to PT makes it more selective because it is lowering the

concentration at other, less-occupied sites.

Can the authors more directly test how many drug molecules are bound? Is multiple binding necessary for splicing effect?

Would cooperativity with an ASO mean there's another splice site? What are the possible mechanisms of cooperativity while only binding one site?

If branaplam and risdiplam bind the same site then how can ris and ASOi6 be synergistic but not bran and ASOi6? Given that they saw so many null results using the Hill coefficient in this experiment, how can they then say with confidence that risdiplam doesn't bind two sites?

All the models analyzed are fully thermodynamic, yet the authors note that there can be kinetic complexities. It seems important to consider what conclusions would be affected by this other class of models. E.g. if the data aren't thermodynamic then can one make conclusions using it about the mode of binding and comparisons to NMR experiments?

Additional Points:

The reproducibility of biological replicates for RNA-seq experiments and relevant statistics (linear fit parameters, r values, RMSE, etc.) must be reported.

Where p values are reported, the authors must specify the statistical test or method for calculating them used.

The authors should briefly describe methods for how their MPSA libraries were cloned and validated, provide coverage statistics for their pooled plasmid libraries, and describe how they determined if the barcode had an effect on the 5'ss of interest.

Values for all fit parameters and their errors should be reported (values are not reported for E_{max} , E_{2x} , weights W_x , or for S).

The multi-drug effects were suggested to exhibit synergy based on 2-3 fold effects. Given the complexity of the system and ad hoc nature of the analysis (especially the variable Hill coefficient) one might want larger effects to be convinced or to interpret these results.

FIGURE 2E: It is unclear to me how you can fit the plots to the far right with any confidence in your model. I also ask that the authors comment on the outliers they observe, especially on the far left pair of plots. What are the physical causes of drug insensitivity? Do these represent sites that cannot be bound by the drug for some reason or areas where U1 binding is not rate-limiting for splicing, perhaps? Do they share any common sequence features?

FIGURE 3: It is not clear in G and H what $\Delta \log W_x$ is in reference to. In the SI, it is explained that it is in reference to the mean for that position. This is difficult to find and difficult to interpret. I would suggest instead picking from the min motifs a consensus base at each position and using that as the referent for $\Delta \log W_x$. By doing this, it becomes clear how mutations lead to free energy penalties and bonuses that can be readily added in an additive model to determine the full effect and what the effect size of each sequence variant is.

The splicing inhibitors are added for 48 hours when the library is transfected. During that time, the off-target effects of the compounds may cause significant changes in the biology of the system, as seen with the changes to splicing of cell cycle, metabolic, and RNA metabolism transcripts in a recent study that performed 24 hour treatments (Ottesen et al., 2023, NAR). It seems important to do control experiments, for example for different incubation times for a subset of constructs to confirm that the cellular response to the drug is not a confounding factor in the results presented here.

Are there DMSO effects? The authors appropriately compare results of drugs added in DMSO to DMSO alone, but it would be of interest to know if there is an effect of this amount of DMSO alone.

Also, as noted above, the drug is present for an extended period of time; is there evidence for or against significant metabolism of the drug over this time?

What fraction of data was thrown out for "alternative splicing events" in each sample? Are there trends observed and are these sequences available for others to analyze?

Figures 1D,E,H,I all have a point that is differently colored at graph coordinates (10,100). What is the significance of this?

Figure S2A-E. An analogous graph with all sequences with consensus base (for S2A, -4A as an example) would be helpful.

I ask the authors to clarify the method of Bayesian parameter fitting. Is this a maximum a posteriori estimate, from MCMC sampling, etc.?

“Allelic manifold” seems like a needlessly complicated and potentially misleading term. A manifold is an abstract topological space with locally Euclidean behavior, where “locally Euclidean” is defined by the preservation of certain continuity, differentiability, or geometric properties under invertible maps from open subsets of the manifold to subsets of Euclidean space (Lee, 2012, Intro. to Smooth Manifolds, Springer GTM). While the graph of any smooth function is a smooth manifold, the manuscript neither needs nor uses any of the mathematical or conceptual technology of manifold theory to do its comparatively-simple statistical mechanical analyses, nor is there any focus on specific allelic differences/variants. I would suggest instead referring to this as a “statistical mechanical model” or “phenomenological model”. (I note that a previously published paper used the “allelic manifold” terminology [Forcier et al., 2018, eLife]; in that paper’s published reviews, some reviewers also found this term to be confusing.)

An interactive process is used to redefine the hyp motif based on MPSA and genome-wide data—and the authors acknowledge the models are still qualitative at this point.

Line 292: “bran motif” should be changed to “hyp motif”.

The logo motif vs. NMR structure comparison may not be as straightforward or conclusive as suggested by the authors. The NMR study doesn’t have the same bases but it still binds the drug. If each tautomeric form of branaplam binds differently, then it would be (highly) unlikely that they are energetically equivalent. The MD simulation (shown for one tautomer) seems superfluous. If there are more rotatable bonds there’s more conformational entropy and chances to bind in different conformations.

Why use EC2x instead of EC50? It is more difficult to interpret EC2x. In the supplemental information, the authors mention that they chose the drug concentrations from pilot dose-response experiments where they measured EC2x. They should show the data from these experiments in a supplemental figure and define what the EC2x is, how it was determined, and in what context it was measured, as well as provide errors on these parameters. When concentrations 7.1x and 10x above the EC2x were chosen for each drug, is this within the plateau of the dose-response regime? As a control, I suggest that the authors also perform experiments using other drug concentrations to rule out differences arising from different treatment concentrations.

Line 484: Kinetic proofreading requires net energy input (usually ATP hydrolysis; Ninio, 1975, Biochemie; Hopfield, 1974, PNAS). Where would that come from?

I follow the logic of validating known targets of these drugs, but with a genome wide screen why not go a step further to describe off targets, new targets, etc.?

Along those lines, we suggest that the authors comment on and compare their work to the recent similar study by Ottesen et al. (2023) in Nucleic Acids Research. In that study, the authors used RNA-seq to measure the off-target effects of branaplam and risdiplam transcriptome-wide and with mini-gene reporters. Does the model in this study predict those effects and their sizes? How do transcriptome-wide data from this study compare?

Reviewer #3 (Remarks to the Author):

Ishigami, Wong et al report a biophysical model to quantitatively define the titration effects and sequence specificity of the 5’ splice site-modifying drugs risdiplam and branaplam, which are currently used (risdiplam) or proposed to be used (branaplam) as therapies for Spinal Muscular Atrophy, and in the case of branaplam also as a potential therapy for Parkinson’s disease. Their results provide molecular models to explain the distinct sequence specificities of these drugs, arguing that branaplam (but not risdiplam -contrary to previous proposals-) uses two distinct interaction modes to recognize 5’ splice sites. In addition, the authors report cooperativity and synergy effects between these drugs and splice-switching antisense oligonucleotide drugs.

Quantitative modeling of the effects of splicing-modifying drugs will be key to develop rigorous pharmacological understanding of the mechanisms of action of these compounds and rationally design drugs of improved activity and specificity. The work by Ishigami, Wong et al provides a very valuable framework to approach these goals. Among other insights, the results reveal sequence specificity determinants for drug specificity that go beyond the “bulge repair” mechanism inferred from high resolution structural data, a popular model whose appealing simplicity is challenged by the new results of Ishigami, Wong et al. The paper reports an important volume of work and an original combination of high-throughput mutagenesis and transcriptome data, biophysical modeling and careful drug titration experiments.

In my opinion the following revisions would help to improve the manuscript:

1) A general comment is that at various points the manuscript is challenging to follow and therefore would benefit from additional explanations and clarity. For example, the text concerning panels 1D-1I soon becomes a bit cryptic and would benefit from additional explanations of the exact meaning of “ris min”, “ris max”, “hyp min” and “hyp max”. Similarly, the text can become cryptic for non-specialists when describing the validation of the biophysical modelling efforts or the falsification of the two-site hypothesis for risdiplam. The final messages are clear, but the narrative leading to them could be streamlined.

2) Figure 1D/E: i) a substantial number of “ris min match” and -specially- “ris max match” are located in the diagonal and therefore they show no much difference in PSI values between DMSO and risdiplam, i.e. a number of 5' ss matching the motifs are not really affected by the drug, arguing that the presence of the motif is insufficient to confer sensitivity to the drug at low or high PSI values. This is easier to explain at high PSI values, as there is little room to see enhanced exon inclusion, but more difficult to explain at low PSI values, where there is ample margin for detection of improved inclusion levels. Related to this latter point: a substantial number of measurements appear to fall in the range between 1% and 0.1% exon inclusion, are these differences reliable? (see also similar points below); ii) The presentation of the data on the the effects of branaplam against DMSO is confusing, because they are shown (albeit with a different format) in Figure S3 (by the way, Figure S3 is labeled as Figure S5 in the figure itself) and again in Figures 1H and 1I, but without an associated motif. The Figure may be easier to follow if panels H and I are swapped with F and G, such that the effects of individual drugs are seen first, and then the comparison of the differential effects between the two drugs; iii) It is not straightforward to conclude, from Figure S3 alone, “that the minimally permissive IUPAC motif that matches all 5'ss in class 1 also matches some 5'ss in class 2”; iv) I am not convinced, at this point in the paper, that this conclusion is granted: “These observations are consistent with a “two-motif” model for branaplam, i.e., one in which branaplam recognizes two distinct but overlapping classes of 5'ss: 5'ss also activated by risdiplam, and 5'ss hyper-activated by branaplam relative to risdiplam.” Why should the results reported at this point of the paper necessarily invoke a “two-motif” model, rather than simply lower sequence specificity requirements for branaplam?

3) Figure 2E: “We note, however, that some data points deviate significantly from their corresponding inferred allelic manifolds; this indicates that genetic context outside of the 10-nt 5'ss sequence has a detectable (if secondary) influence on drug effect. Understanding the mechanistic basis for this secondary influence on drug effect is a worthy goal for future research.” The two examples that differ significantly from inferred allelic manifolds (SMN2 and FOXM1) are those in the diagonal of panel 4D, while the two that fit nicely the curves (SF3B3 and HTT) show much higher E values for branaplam than risdiplam. Does this say anything about the model being far more accurate to predict sequence determinants that determine hyperactivation by branaplam, but less so for common sensitivity to risdiplam / branaplam? Can this be used to further refine the model?

4) Figure 3: the authors conclude that “Plotting these data against model predictions confirms that the two interaction-mode model for branaplam explains these data for both drugs well (Fig. 3C-F) and better than the one-interaction-mode model for branaplam does (Fig. S12).” R2 values for the two-

interaction mode in panels C-F are really not so different from those of the one-interaction mode shown in S12: 0.400 vs 0.403, 0.986 vs 0.986, 0.634 vs 0.601 and 0.960 vs 0.939. Based on these data alone, the two-interaction mode seems to be only marginally statistically better than the one-interaction model.

5) The statement "One prediction of the two-site hypothesis for risdiplam is that a molecule of risdiplam bound to the PT should promote SMN2 exon 7 inclusion even when a second molecule of risdiplam is prevented from binding to the U1/5'ss complex." may not necessarily reflect the prediction of the two-site hypothesis (as far as I understand it): the two-site hypothesis may require concerted actions from the two sites to effectively induce exon skipping, one hit not being sufficient for achieving detectable effects.

6) Figure 5: it would be good to discuss how the determination of Hill coefficients may be affected by the intrinsic inaccuracy of measurements of inclusion/skipping ratios, in particular with ratios above 10 or below 0.1 (e.g. a significant number of measurements are between 0.1 and 0.01 or even below: are measurements below 5% of exon inclusion accurate, reproducible and trustable)? In panels 5I-L branaplam is wrongly spelled (branplam).

7) Figure 6: as for the previous point, a similar question of accuracy in the determination of inclusion values above 10 or below 0.1 could be made. This is particularly relevant for panels L to O, in which the measurements range from 90-91% inclusion to 93-96% inclusion.

8) The synergy effects of Figure 6 may not be easy to follow because the effects of risdiplam / branaplam are not in the same figure, so one needs to jump between figures to compare Hill coefficients and it is not clear what comparisons correspond to what statistical significance values. It would be great to have a Table in which the comparison between Hill coefficients would be straightforward to follow.

9) The sentence "These findings definitively establish that a splice-modifying drug can exhibit cooperativity even when it binds only a single site on target pre-mRNA, and that the extent of this cooperativity can differ between drugs that promote inclusion of the same cassette exon." assumes that drugs bind to a single site on target pre-mRNA, but this is not necessarily the case, particularly when using ASOs that may have a single target with 100% complementarity, but that may engage with other sites, even if not fully complementary.

Reviewer #4 (Remarks to the Author):

The problem is certainly relevant, and a solid clarification of the mode of action (MOA) of these 2 drugs would be highly informative and impactful. However, the work is quite complex and hard to follow for someone who is not too familiar with this kind of parallel splicing assays.

Reading the paper, I got easily lost. For example, the first paragraph of the results is more about the methodological approach, rather than 'results'. The overall indication of the MOA is potentially interesting, but all these indirect observations still leave quite some room for potential interpretations that may differ from those reported.

There are several instances in which drug cooperativity and multi-drugs synergy is used to argue about drug efficacy and MOA. I admit my limitations here, but may the overall story could be made easier to follow and more efficient in describing what would be the ultimate message - is it about a massive experimental framework for characterizing RNA-targeted small molecules or is it about the

ultimate clarification of the MOA of risdiplam and branaplam ?

Other drugs are mentioned, like RECTAS and BPN-154. What about these? Are not included because this framework is too costly?too much time-consuming?

I understand the paper wants to use this experimental framework on those two drugs, but I found it confusing (to me).

Response to reviewer comments

Overview

We greatly appreciate the referees' time and effort in providing feedback. In response, we have performed additional experiments and made extensive revisions to the manuscript. In particular:

1. We have simplified the IUPAC motif analysis presented in **Fig. 1** and **Fig. 2**, as well as the presentation of the biophysical modeling analysis in **Fig. 3**. These changes should address concerns about the clarity of these analyses.
2. We have simplified the discussion of how our data contradicts the prevailing two-site hypothesis for risdiplam.
3. We have performed additional experiments to assess the off-target effects of the splice-modifying drugs (**Fig. S12**). We find that none of the drugs in our study have substantial off-target effects. These data should address concerns that off-target sites for these drugs might explain our observations concerning anomalous cooperativity.
4. We have performed additional experiments to assess multi-drug synergy. The new data are presented in a simplified **Fig. 6**, and directly demonstrate multi-drug synergy for all assayed pairs of splice-modifying drugs (except for risdiplam and branaplam, as expected).

We believe the revised manuscript is substantially improved as a result of these new data and revisions. We believe these changes, together with our point-by-point response to the referees' critiques, will address all of the referees' core concerns.

In what follows, referee comments are marked in *black italic text*, while our point-by-point responses to the referee comments are marked in **blue regular text** and preceded by a ► character. Relevant references are listed at the end of the document.

Response to Reviewer #1

Reviewer #1 (Remarks to the Author):

The authors of this paper present an interesting approach based on RNA sequencing and data modeling to understand the sequence basis for the mechanism of action of two small molecule drugs Risdiplam and Branaplam. This is an intensive field of research considering the few small molecules that act on RNA. Several modes of action have been proposed based on structural work and one mutagenesis but a complete understanding of the mode of action and in particular why so few alternative exons are targeted by these molecules that deeper understanding is needed. Here the authors have elucidated the sequence motifs that are important for the enhancing splicing activity of the two molecules. They also discover two sequence motifs for branaplam. This sequence motif analysis confirms the importance of a bulge adenine in position -1 but also other key residues in the 5' splice-site sequence. This suggests that they're more than a -1 A bulge. Moreover, they could show evidence for a synergy between several molecules, cooperativity and also that the purine-rich region upstream of the 5'SS may not be a binding site for these small molecules. Overall, this is a very important study that gives some clues on the sequence-specific needs for the small molecules to act.

► **We thank the reviewer for this positive assessment.**

I do not have any major criticisms. Maybe the sequence motifs found could be discussed a little bit more.

1. *For example the -2G maybe important to allow the formation of the bulge A.* ► **This is a good suggestion. We have revised the Main Text to state "[The] A₋₁ bulge is stabilized by the 5'ss G₋₂ pairing in a shifted register with C₉ of the U1 snRNA" in the discussion of Fig. 4A.**
2. *The -3A can base-pair with the opposite U of u1snRNA and this might explain why this is needed. This would mean that branaplam requires a more stable helix. Same for the preference for A/G in position +3, this would favor pairing for a pseudo U of u1snRNA. I think this is worth commenting on in the discussion.* ► **This is a good suggestion. We have revised the Main Text to state "For example, we find that the hyper-activation energy motif (for branaplam) strongly favors an A at position -3 (which**

might base-pair with U₁₀ of the U1 snRNA) and that the risdiplam energy motif favors an A or G at position +3 (which might base-pair with Ψ₆ of the U1 snRNA).” in the Discussion.

Response to Reviewer #2

Reviewer #2 (Remarks to the Author):

In this paper, Ishigami et al. develop a quasi-biophysical model to quantitatively describe the influence of splice-modulating drugs (risdiplam, branaplam, and ASOs) on splicing. They lay out models that relate U1 snRNP and drug occupancy to observed splicing outcomes (PSI) in an effort to quantitatively describe the drug specificity. They use these models to propose, interpret, and evaluate models for drug binding and action. This level of quantification of drug effects is generally missing from the field and reflects a shift back to biochemical and biophysical fundamentals in the modern era of cellular studies and high-throughput sequencing that will be necessary to move the field forward.

Overall the approach is an important—even critical—one, which others will likely follow in studying splicing and other aspects of RNA metabolism.

► We thank the reviewer for this favorable assessment. The reviewer has provided a very thorough review of the paper, so here we briefly summarize the major points that were raised and how we have addressed them.

- (1) The primary concern expressed by the reviewer was that the quantitative models presented in the manuscript were not described with sufficient clarity and precision. In response, we have clarified and made more precise the description of the quantitative models and the assumptions that underlie them.
 - a. The IUPAC motifs identified in **Fig. 1** are now explicitly described as resulting from an “exploratory analysis”. This exploratory analysis has also been simplified. The results of this exploratory analysis were used only to motivate the mathematical form of the biophysical model in **Fig. 3**; the identified IUPAC motifs (in **Fig. 1**) are not used in either the definition of the model or the inference of model parameters.
 - b. The allelic manifold model, used to infer drug effect values in **Fig. 2**, is now presented as an explicit biophysical model in **Fig. 2**, **Fig. S5**, and **Supplemental Information (SI) Sec. 3.1**
 - c. The biophysical model of **Fig. 3**, as well as of **Fig. S6**, are now written explicitly in terms of states and Gibbs free energies, rather than weights. **SI Secs. 3.2** and **3.3** give detailed mathematical definitions of these models.
 - d. The dose-response model used in **Fig. 5** and **Fig. 6** is now explicitly described as an “empirical” model, rather than a “biophysical” model, as it requires non-physical assumptions. These issues are discussed extensively in **SI Sec. 3.4**.
- (2) The reviewer expressed concern about the clarity of the methods and results. We have substantially expanded the **SI** in response, and added numbered sections that are now referenced in the paper.
 - a. **Table S1** now shows inferred parameter values for the dose-response curves in **Fig. 5**, **Fig. 6**, and **Fig. S13**.
 - b. **SI Sec. 2.1** now gives the explicit mathematical formula used to quantify PSI from MPSA data.
 - c. **Fig. S1** and **Fig. S2** have been augmented to show complexity and coverage statistics for the MPSA libraries.
 - d. **Fig. S1** has been augmented to show pilot data and analysis results from which 1x risdiplam and 1x branaplam concentrations were determined.
- (3) The reviewer expressed uncertainty about the interpretation of the data. In response we have performed additional experiments and analyses.
 - a. **Fig. 6** presents new linear mixture data that shows that all combinations of drugs assayed (except for the risdiplam/branaplam mixture, as expected) show statistically significant synergy as assed by linear mixture curves. These new results explicitly demonstrate widespread multi-drug synergy. The Hill coefficient comparison tests for synergy are less direct and have been moved to **Fig. S13**.

- b. **Fig. S11** now shows, via dose-response curves on non-target minigenes, that none of the drugs assayed have significant off-target effects on splicing in general.
- c. **Fig. S14** has been added to show probability logos for 5'ss detected and/or activated in RNA-seq data. A discussion of these logos is provided in the caption of **Fig. S14**.
- d. We confirmed by BLASTN that ASOi7 does not have significant off-target complementarity in the SMN2 minigene, which supports our conclusion that anomalous cooperativity is not caused by off-target sites.

Nevertheless, there are major limitations in presentation, data, analysis and interpretation:

1. *The models are not clearly presented. As I understand it the authors are creating a logo or position weight matrix in a two step process—what sequences determine splice site choice and then what sequences determine whether drug binding will alter the splice site choice. They then create a binding model based on an assumed independence of the terms in the sequence logo, trying multiple versions of defining those logos. If the above description is accurate, the authors should clearly lay this out, along with the assumptions. If it is not accurate, a clearer description is needed, at least for this reviewer. Note that the assignment of logo sequences seems qualitative and thus to an extent arbitrary. This is a significant limitation that minimally needs to be clearly laid out.* ► We agree that the presentation of our modeling strategy for drug specificity was unclear. We have substantially revised the manuscript to address this concern. The revised manuscript now explicitly states that our modeling strategy has two parts. The first part is an exploratory analysis in which we identified qualitative IUPAC motifs for the 5'ss specificities of risdiplam and branaplam that were consistent with MPSA and RNA-seq data. The second part is a biophysical modeling analysis in which the results of the exploratory analysis were used to motivate a quantitative thermodynamic model for the 5'ss specificities of risdiplam and branaplam. In this second part, the parameters of the biophysical model, including of the 5'ss specificity motifs, were inferred from MPSA and RNA-seq data using Bayesian inference. We have clarified that the thermodynamic model does indeed assume that each nucleotide within the 5'ss contributes independently to the Gibbs free energy of each drug binding to the U1/5'ss complex. These nucleotide-specific energy contributions define the “risdiplam energy motif” and the “hyper-activation energy motif” (reported in **Fig. 3F,G** of the revised manuscript).
2. *In addition, the models presented need to show the splicing event and how that is affected by U1 and drug binding. I think there are unstated assumptions in the model (beyond site independence) and the authors should lay out a full molecular model and then derive an expression from the model. I.e., directly connect the math to the model that is presented.* ► We have clarified the assumptions that underlie our biophysical models for splicing. Briefly, all biophysical models are thermodynamic models that assume PSI is 100 times U1/5'ss occupancy in thermal equilibrium. These are all highly simplified models of the splicing process, but they makes sense if E complex formation is limiting for splicing. This model is also motivated by the success of similar thermodynamic models in studies of transcriptional regulation. These assumptions are now laid out explicitly in the **Main Text**, in **SI Sec. 3**, as well as in **Fig. 2B**, **Fig. 3**, **Fig. S5**, **Fig. S6**, and **Fig. S11**.
3. *The analysis allows a non-unit Hill slope, but the presented biophysical models do not. If the Hill slope is empirical, then the model becomes empirical, not biophysical. Further, showing that a biophysical model does not hold is valuable and then proposing (as was done later in the manuscript) why a model may not hold is also valuable. There more can be done to actually make the models and show what is needed from the revised models to account for the data.* ► We agree with this assessment in regards to the concentration-dependent drug-effect model (**Fig. 5C**, **Fig. S11**, and **SI Sec. 3.4**). [Note that this critique does not apply to the drug-effect model (**Fig. 2B**, **Fig. S5**, **SI Sec. 3.1**), the two-interaction-mode model (**Fig. 3A**, **SI Sec. 3.2**), or the one-interaction-mode model (**Fig. S6**, **SI Sec. 3.3**), all of which are genuine biophysical models]. We now explicitly describe the concentration-dependent drug-effect model as an “empirical” model, not a “biophysical” model. Moreover, we explain in **Fig. S11** and in **SI Sec. 3.4** text that, while the mathematical form of the empirical model has a biophysical motivation, the model itself does not have a rigorous biophysical interpretation because it requires two non-physical assumptions: (i) that H drug molecules bind simultaneously (where H need not be an integer), and (ii), that there is a 5'ss state in which drug is bound in the absence of bound U1. Nevertheless, our results show that the empirical concentration-dependent drug-effect model is still

useful for discerning drug mechanism, as it allows one to discern from dose-response curves whether mutations in a minigene impact drug mechanism (i.e., change the value EC_{2x} or H) or simply impact context strength (i.e., change the value of S).

4. *The data themselves are not presented with clarity and thoroughness needed to evaluate to what extent limitations in the model vs. in the data are responsible for deviations, and it is unclear at least to me what errors or deviations should be taken as invalidating the model vs. being within experimental error (either random or systematic). (I.e., what error models are used, in detail?) In particular the absence of saturation in most drug 'binding' curves need to be made clear by presenting sufficient underlying data. Also, actual PSI values rather than normalized (as described in prior papers referenced for the method) should be given and the reader should be able to connect these values to the number of sequence reads and error estimates from all sources.* ► The referee raises multiple good points which we now address in turn. (1) The revised **SI Sec. 2.1** now explicitly describes how, in our MPSAs, PSI values were calculated from sequencing data. We note that this procedure yields enrichment ratios, not absolute PSI values. These enrichment ratios are converted to PSI by normalizing by the median of the enrichment ratios observed for four consensus splice sites. We have included radioactive gel data in the revised **Fig. S1**, which establishes that the four consensus splice sites all have PSI of 100, and thus that this normalization procedure is valid. (2) All Bayesian models and inference procedures, including the inference for the allelic manifold (**Fig. 2**), the sequence-dependent drug models (**Fig. 3**, **Fig. S6**, **Fig. S7**), the dose-response curves (**Fig. 5**, **Fig. 6**, **Fig. S12**, **Fig. S13**), and the linear-mixture modeling curves (**Fig. 6**), are formally described in **SI Sec. 4**. All of the Bayesian models also include explicit error models. For the allelic manifold model and the sequence-dependent models, we generally do observe that the Bayesian inferred errors are larger than the experimental uncertainty, as quantified by the variation between biological replicates, thus indicating a detectable level of model mis-specification. Such signatures of mis-specification, however, are typical in the quantitative modeling of biological data, and in this case does not invalidate the utility of our models. (3) All data used to infer each dose-response curve are shown in the same plot as that curve. The numerical values of the dose-response-curve parameters S , E_{\max} , EC_{2x} , and H , are now reported (as medians and 95% credible intervals) in **Table S1**. The results show that the parameters S , EC_{2x} , and H can all be determined to high precision even in the absence of saturation in the dose-response data; only the parameter E_{\max} requires that saturation be observed for a high-precision value to be inferred. Because of drug toxicity issues, it was not possible to measure drug concentrations at which many of our dose-response curves saturate. As a result, many of our dose-response curves do have highly uncertain values for the parameter E_{\max} . This is not a problem, however, because none of our claims rely on the values of E_{\max} .
5. *I appreciate the efforts to lay out quantitative models laid out here, yet the conclusions drawn from those models need further tests before publication and the manuscript as a whole requires further clarification. In particular the models are not clearly laid out, include features and empirical assumptions beyond the biophysical, and the presentation is unclear in several places.* ► We agree that the assumptions underlying the quantitative models were not clearly laid out in the original manuscript. The revised manuscript presents the assumptions underlying these models much more clearly. We also made major revisions to improve the overall clarity of the manuscript. We believe this improved clarity will address the referee's primary concerns. To provide additional tests of our claims of synergy between splice-modifying drugs, we measured additional linear-mixture curves (**Fig. 6H,I,K**). We also performed additional control experiments (**Fig. S1I**, **Fig. S12A-F**).

In addition to the above, I suggest the following major and minor points that I believe will strengthen the paper overall.

Major Points:

6. *Generally, the authors claim to have attained quantitative definition of sequence-dependent splicing, understanding of the mechanism of drug action, and novel strategies to apply therapeutically. It is not clear to me that these claims are met. What is the definition of "quantitatively defined" and what is the metric?* ► This is a fair critique. We have replaced "quantitatively defined" with "quantitatively characterized" to make it clear that we do not believe our quantitative models provide a final definitive description of the 5'ss specificities of risdiplam and branaplam.

7. *The data appear to give Hill slopes $\neq 1$ but the mechanisms in the presented biophysical models do not predict that.* ► This is a fair criticism. We now describe our model for concentration-dependent drug effect (**Fig. 5C,D**) as an “empirical” model instead of a “biophysical” model. The mathematical form of this model is motivated by an explicit biophysical model, which is illustrated in **Fig. S11** and described in **SI Sec. 3.4**. But as discussed in **SI Sec. 3.4**, this model and the inferred dose-response curves do exhibit some biophysically implausible behavior (non-integer Hill coefficient and drug bound in the absence of U1). Potential mechanisms for this non-biophysical behavior are given in the **Discussion** and in **SI Sec. 3.4**.
8. *MPSA and motif description: Line 87-88: The goal of finding the sequence features of a functional 5'ss and separating that from the sequence features necessary for drug binding is central to this study. This point could be made clearer by including the logo motif from the DMSO control and then stating what bases change in the motif for drug binding.* ► We have added **Fig. S14** and referenced this figure in **Discussion**. **Fig. S14** shows probability logos for 5'ss active in DMSO, 5'ss active under treatment with risdiplam or branaplam, and 5'ss sensitive to risdiplam or branaplam, as determined from RNA-seq data. The corresponding caption compares these logos to the interaction-mode-specific IUPAC motifs (**Fig. 1**) and energy motifs in (**Fig. 3**). We note that the IUPAC motifs in **Fig. 1** are discriminatory motifs, and it would not be possible to determine an analogous motif from the DMSO data alone. Similarly, the logos in **Fig. 3** are inferred through a Bayesian procedure that accounts for all the data, including the DMSO data, and it would not be possible to infer an analogous motif from the DMSO data alone.
9. *The criteria for classifying the 5'ss that are activated by drug, insensitive, or undeterminable seem arbitrary as presented in the methods. How were the regions defined? What happens as those criteria are shifted?* ► In the revised manuscript we clarify that the analysis to determine IUPAC motifs was exploratory, and is used only to motivate the hypotheses that we formally evaluate in **Fig. 3**, **Fig. S6**, and **Fig. S7**. In the revised manuscript we also clarify that the region boundaries (formally defined in **SI Sec. 2.2**) were chosen to distinguish the qualitatively different classes of sequences we observed in these initial experiments. For the case of risdiplam vs. DMSO and branaplam vs. risdiplam, there was indeed some arbitrariness in the specific region boundaries chosen. This arbitrariness is legitimate because we only claim that some IUPAC motifs are consistent with some reasonable choice of region boundary. For the case of branaplam vs. DMSO, we claim that there is no IUPAC motif that is consistent with any reasonable choice of region boundary. This claim is supported in the revised manuscript through a robustness analysis in which a range of plausible region boundaries are tested (**Fig. S4**).
10. *To determine the hyp motifs, a ratio of $PSI_{\text{branaplam}}/PSI_{\text{risdiplam}}$ was used. I suggest considering including a plot of the $risdiplam/branaplam$ ratio.* ► We chose to plot PSI_{bran} vs. PSI_{ris} to keep the presentation consistent with the PSI_{ris} vs. PSI_{DMSO} plot and the PSI_{bran} vs. PSI_{DMSO} plot. We also prefer to directly plot PSI values, as these values more directly reflect the MPSA data. We are reluctant to also include a plot of $PSI_{\text{bran}}/PSI_{\text{ris}}$ vs. PSI_{ris} , because the information provided in that plot would not be meaningfully different than the information provided in the PSI_{bran} vs. PSI_{ris} plot.
11. *In Figure 1D-I, there are several sequences that match the motif but are in a region with little to no splicing effect in the DMSO control or with drug treatment (gray cloud). I suggest that the authors elaborate on this as it could also be related to their point about a functional 5'ss and be used to further refine the model.* ► In the revised **Fig. 1D-E**, **Fig. S3**, and **Fig. S4** there are 5'ss sequences with activities below the lower PSI threshold, both with and without drug. For these 5'ss, we do not observe stimulation under drug treatment. This could be due to two different reasons: drug does not affect those 5'ss; or those 5'ss have a context strength so low that the effect of drug is not sufficient to raise PSI above the threshold. Similarly, there are 5'ss sequences with activities above the upper PSI threshold, both with and without drug. For these 5'ss, we do not observe stimulation under drug treatment. This could be due to two reasons: drug does not affect those 5'ss; or those 5'ss have a context strength so high that the effect of drug is not detectable. We therefore excluded both classes of 5'ss sequences when searching for IUPAC motifs. The revised text makes this point clearer.
12. *I'm not sure I follow the logic of defining min and max motifs. It seems from the MPSA that the min motifs are sufficient to describe any data points classified as having an effect. Moreover, the authors derived the motifs with different algorithms. I think one can consign max motifs to SI and reduce the*

discussion thereof, focusing on the min motifs as starting points for the subsequent experiments. Most fundamentally, defining these motifs is arbitrary and provides the authors multiple 'shots on goal' to come up with a model that fits. Stated another way, a model based on a motif is not really biophysical unless the motif is used to define affinities; it seems that this is (sort of—see below) used to quantitatively predict affinities and then splicing outcome but one needs to dig through the model details to figure this out (see point 2 below). ► We agree that the presentation of the min and max motifs was unnecessarily complex and obscured the exploratory nature of the analysis in **Fig 1**. The relevant text has been rewritten to make the logic of this part of the paper clearer. Briefly, the IUPAC motif analysis of the MPSA data was exploratory, as our only goal was to find some IUPAC motifs that could plausibly explain drug effect. We find that there are multiple risdiplam motifs and hyper-activation motifs that do this. The min and max motifs simply provide bounds on these possible motifs and are now relegated to **Fig. S3**. We also find that there are not any branaplam motifs that plausibly explain drug effect, as demonstrated in **Fig. S4**. This exploratory analysis thus motivates the two-interaction-mode hypothesis for branaplam, and specifically, the mathematical form of the biophysical model described in **Fig. 3**.

13. As presented, it is highly unclear how PSI is determined. The authors must clarify exactly what they did in the Methods; merely referring to another publication is insufficient. ► We agree. The revised **SI Sec. 2.1** now provides equations that explicitly show how PSI was determined from read counts in the MPSA experiments (**Eqs. 2-7**).
14. Presentation (see also below): Motifs are used to make predictions but the logic and workflow is not defined and the reader needs to read the Methods (supplement) to figure this out. That said, there also appear to be arbitrary cut-offs used to define different levels of motifs—that are later used to see which gives better fits? If this is not the case, the authors should clarify. If it is, they should present metrics for the agreement of the data and model using each cut-off or some better way to define the logo, and thus residue weighting, in the model. ► We agree that our logic was unclearly presented. The revised text makes clear that the inference of IUPAC motifs was exploratory. We believe this will address the referee's concerns.
15. Coarse-grained biophysical model. The biophysical models are partial. States and weights are defined, but the partitioning between splice products is not shown in the models. I think I figured this out, but it should be shown. ► Indeed, this was not clear. The revised text clarifies that PSI is assumed to be equal to the equilibrium occupancy of U1 on 5'ss pre-mRNA. Moreover, all relevant states of the biophysical model are now presented in **Fig. 3A**.
16. Lines 263-265: It is not an assumption of the model that each state has a weight, but just how Boltzmann statistics work to describe any system at thermodynamic equilibrium, which is what seems to actually be assumed. I suggest revising to clarify. ► The presentation of the biophysical models has been revised to explicitly represent the Gibbs free energy of each possible state, rather than the corresponding Boltzmann weight. We believe this explicit discussion of Gibbs free energies will clarify the presentation for the reader.
17. Lines 268-269: What the authors seem to be saying is that the energetic contributions ($= -kBT \ln W_x$, where W_x is a Boltzmann weight) of a given 5'ss to drug/U1 binding are the sum of the energetic contributions of the constituent nucleotides. I think this is a clearer and more interpretable way of stating things as opposed to "the logarithms of the weights... are taken to be additive functions of the 5'ss sequence". One may also point out that energetic additivity is a common, if implicit, underlying assumption of sequence motif descriptions of macromolecule binding. See Jarmoskaite et al. (2019), *Mol. Cell* for similar statements. ► We agree. The revised presentation of the biophysical model for 5'ss sequence specificity now states that each nucleotide in the 5'ss is assumed to contribute additively to the Gibbs free energy of drug binding to the U1/5'ss complex. The revised text notes that additivity assumptions like this are common in thermodynamic models that describe sequence-dependent interaction energies.
18. I suggest clarifying the models by combining them where applicable. In Figure 2A, for example, there is actually only one model, but the case without the drug has $E = 1$ (i.e., as all weights $W_x = 0$ identically since $[drug] = 0$). ► The revised text defines the biophysical models more clearly. In **Fig. 2B** (revised from **Fig. 2A** of the original manuscript), we still choose to show PSI with and without drug, because

those are the quantities plotted in **Fig. 2C** and **Fig. 2E**. The revised **Fig. 2B** caption clarifies that $E = 1$ when $[\text{drug}] = 0$.

19. *In general, I think it would clarify the text and improve interpretations to be more explicit about what these weights are in terms of free energies and drug concentrations, which are easier to understand/interpret, and more accessible to a general biology/biochemistry audience.* ► We agree. In the revised text, all biophysical models are defined explicitly in terms of Gibbs free energies rather than weights.
20. *In this framework, S should be a function of the free concentration of U1 snRNP and its affinity for the site. Assuming the former stays the same, do fit values for S match what we know about U1 binding preferences? Do they teach us anything about these?* ► This is a difficult question to answer, because S , which quantifies exon “context strength”, depends not just on 5’ss sequence, but also on other factors, including exon size, 3’ss strength, branch point strength, the RBP binding content of the exon and proximal intron, etc.. We believe that understanding how each of these factors contributes to S would indeed be valuable, but would require substantial new analysis and is beyond the scope of the current manuscript.
21. *Why are branaplam and risdiplam given the same weight for the risdiplam interaction mode when they are used at different concentrations and may have different affinities for the same interaction site? What happens if this assumption is shifted?* ► Preliminary dose-response experiments found that branaplam is about twice as potent as risdiplam at *SMN2* exon 7. The relative concentrations of risdiplam and branaplam used in our MPSA and RNA-seq experiments were chosen to compensate for this difference in potency; this is now stated the **Main Text** and in **SI Secs. 1.4** and **1.6**. The use of the same weight matrix for risdiplam and branaplam in the risdiplam interaction mode was motivated by our exploratory analysis of these data, which found that activation by branaplam can be predicted remarkably well by a combination of the risdiplam IUPAC motif and the hyper-activation IUPAC motif.
22. *Figure 3B: is this for all of the data or bran/ris specific?* ► We have clarified in the **Fig. 3** caption that model parameters were inferred from the PSI values measured by MPSA on cells treated with DMSO, risdiplam, or branaplam (**Fig. 1D-F**), as well as from drug effect values E for risdiplam or branaplam determined by RNA-seq (**Fig. 2D**). The comparison between log likelihood values for the two-interaction-mode and one-interaction-mode models, originally shown in Fig. 3B, has been moved to **Fig. S7**, which also shows log likelihood values stratified by dataset.
23. *Figure 3C-F: Why is R^2 for E so much lower than PSI? It seems like points with low predicted E values have higher variance. I suggest that the authors comment on this.* ► The revised text notes that higher R^2 was obtained on the MPSA data relative to RNA-seq data due, at least in part, to the fact that different sets of 5’ss sequences were assayed by the different methods.
24. *Figure S12A-D has almost identical R^2 values for branaplam: What other evidence suggests that this model is correct?* ► The primary quantitative evidence for the two-interaction-mode model relative to the one-interaction-mode model are the log likelihood ratios shown in the revised **Fig. S7**. These log likelihood ratios provide strong quantitative evidence, considering that the two models have the same number of parameters. Note also that there is no direct mathematical relationship between these log likelihood ratios and R^2 . Rather, R^2 values are shown for completeness and interpretability. We also believe that the reduction of R^2 from 0.634 to 0.602 between **Fig. 3D** and **Fig. S6D**, and the reduction of R^2 from 0.958 to 0.936 between **Fig. 3E** and **Fig. S6E**, are substantial, considering the large number of data points used to compute these R^2 values.
25. *Figure 3G,H: what are the differences between this model and the logo motifs found in Figure 1? I suggest that the authors further elaborate on this.* ► The revised text clarifies that the risdiplam IUPAC motif and hyper-activation IUPAC motif in **Fig. 1** are qualitative in nature, were obtained as part of an exploratory analysis, and were used only to motivate the mathematical form of the biophysical model in **Fig. 3**. The IUPAC motifs shown in **Fig. 1** were not otherwise used to define the two-interaction-mode model of **Fig. 3A**, or used in the inference of the energy motifs shown in the revised **Fig. 3F,G**.
26. *How many total parameters does each model have?* ► **Main Text** and **SI Sec. 3** have been revised to explicitly state the number of model parameters. The allelic manifold model (**Sec. 3.1**) comprises

194,129 parameters: context strength values S for 189,087 different genomic exons, and drug effect values E (one for risdiplam and one for branaplam) for 2,521 distinct 5'ss sequences. This model also includes 9 hyperparameters. The models of 5'ss-dependent drug effect [both the two-interaction-mode model (**Sec. 3.2**) and the one-interaction-mode model (**Sec. 3.3**)] have 351 parameters: 33 parameters comprising the risdiplam energy motif, 33 parameters comprising the branaplam energy motif, and 285 parameters describing ΔG_{U1} values for the 285 variant 5'ss sequences assayed in the MPSA. Both models also have 5 hyperparameters describing experimental noise.

27. *Are they [the risdiplam and branaplam models] truly additive or does this assumption need testing?* ► The residual deviations in **Figs. 3B-E** are not due solely to experimental uncertainty and thus reflect some level of model misspecification. Non-additivity might be one such type of model misspecification. However, our data do not allow us to explore this question further. Still, we believe the risdiplam energy motif and hyper-activation energy motif together provide good first-approximation descriptions of the 5'ss sequence specificities of risdiplam and branaplam.
28. *How do train/test errors compare to the error in the entire dataset?* ► We performed Bayesian inference on the entire dataset. As is standard practice in Bayesian inference, there was no split of the dataset into training data and testing data.
29. *Hill coefficient, cooperativity, and titration. There is no term in the model(s) that predicts a Hill coefficient $\neq 1$, so that using other values turns a model into an empirical fit.* ► We agree. The revised text now describes the concentration-dependent drug-effect model in **Fig. 5** as an empirical model rather than a biophysical model.
30. *The authors do later suggest that there may be proofreading or other factors that give different behavior but do not derive mathematical formulas from such models to show whether (or not) such a model can account for the data and what other properties it would have.* ► We have added a reference to the seminal kinetic proofreading papers by Hopfield and Nino, which mathematically show how kinetic proofreading can lead to anomalous cooperativity, i.e., apparent Hill coefficients greater than 1 resulting from only a single binding site. That said, there are a variety of different possible kinetic proofreading models, and we do not believe our dose-response data are sufficient to distinguish between these possibilities.
31. *I don't fully understand the logic of setting the weight of the drug bound to the RNA alone, A , equal to B/E_{max} . I ask that the authors please clarify and explain in the SI (this also relates to the above main point about more clearly defining biophysical/molecular models that connect to the math).* ► The revised **SI Sec. 3.4** explains that the assumption that drug can bind to 5'ss pre-mRNA in the absence of U1 is required for the model to predict E values that saturate at high drug concentration. The data require a model that is capable of producing such a saturation effect, which is clearly observed for some drugs (e.g., the ASOi7 data shown in **Fig. 6A**). We believe that, in reality, saturation is more likely to arise from inefficiencies in the spliceosome cycle that occur downstream of U1 binding to pre-mRNA (i.e., downstream of E complex formation). However, fitting an empirical model is sufficient for this portion of our argument, and the model we have constructed provides an excellent fit to the data (**Fig. 5E-L, Fig. 6A-D**).
32. *The model predicts that as one titrates the drug, the drug effect E will increase linearly with the concentration of drug until all splice sites are saturated, after which it will not change. Is this the case?* ► E is approximately 1 for sufficiently low drug concentration. Then, starting at a concentration of approximately EC_{2x} , E increases as concentration raised to the power H . E then saturates at a value E_{max} at sufficiently high concentration, due (according to the model) to saturation of drug occupancy.
33. *Are some sites predicted to be already saturated based on E values?* ► Some 5'ss have high context strength, i.e. $S \gg 1$, and so already have PSI near 100. But in the absence of drug, all 5'ss have E values of 1, since E quantifies drug effect.
34. **FIGURE 5: How is E_{max} fit? Is there a specific reason why higher [drug]'s were not included (e.g., toxicity)? (i.e., the data do not appear to saturate in most cases as would be needed to specify E_{max}).** ► The 4 parameters of the concentration-dependent drug-effect model (S , EC_{2x} , E_{max} , H) were inferred simultaneously for each titration curve using the Bayesian inference procedure described in SI. Drug

concentration was varied up until saturation was observed or until drug concentrations became toxic. Toxicity prevented a precise determination of E_{\max} in many cases.

35. *The main data (Figures 5 & 6) are problematic in that an endpoint is not well defined for nearly all the data. While fits often give seemingly good fits, without obtaining a clear plateau, these values can be off by orders of magnitude—and it is not clear how such errors would affect the conclusions.* ► In the dose-response curves for which no plateau is observed, toxicity prevented the use of higher drug concentrations. In such cases, the resulting inferred values of E_{\max} were highly uncertain, but we were nevertheless able to precisely determine values for S , for EC_{2x} , and for H ; see **Table S5**. Note: the inability to confidently determine E_{\max} is one reason we chose to parameterize the curve using EC_{2x} rather than EC_{50} , as now described in **SI Sec. 3.4**. In terms of our scientific conclusions, our results depend only on our measured values for S , EC_{2x} , and H , and do not depend on E_{\max} .
36. *Isn't the fact that disrupting the PT increased the Hill coefficient evidence that it also binds there and just doesn't have the intended effect on splicing? E.g., the effective concentration is higher at the functional site.* ► We do not see how this increase in H provides evidence for the 2-site hypothesis. It is theoretically possible that a second molecule of risdiplam binds to the PT and weakly inhibits (rather than promotes) exon inclusion. That scenario, however, would contradict the published two-site hypothesis for risdiplam. Moreover, the same would have to happen for branaplam, again contradicting the two-site hypothesis for risdiplam. We believe a more likely explanation is that H is affected (via unclear molecular mechanisms) by RBPs that bind near the PT.
37. *Another model is that binding to PT makes it more selective because it is lowering the concentration at other, less-occupied sites.* ► We do not believe the suggested model is plausible. Even if our plasmids were present in thousands of copies per cell, there would still be many more equivalent PTs within the HeLa cell transcriptome, and these genomic PTs would buffer the free concentration of risdiplam and branaplam against changes in the number of plasmid-borne PTs. Also, were transcriptome RNA adsorbing large quantities of drug in our experiments, this would be reflected in a sharp drop in drug effect at lower applied drug concentrations, and that is not what we see.
38. *Can the authors more directly test how many drug molecules are bound? Is multiple binding necessary for splicing effect?* ► We do not see a straight-forward way to more directly test how many drug molecules are bound in cells.
39. *Would cooperativity with an ASO mean there's another splice site? [We believe the referee means ASO binding site here] What are the possible mechanisms of cooperativity while only binding one site?* ► We believe the ASOi7 data conclusively show that cooperativity can obtain even upon binding of only one drug molecule per pre-mRNA. As with small molecule drugs, kinetic proofreading might explain the anomalous cooperativity observed for ASOi7. Possible causes of anomalous cooperativity, including kinetic proofreading, are discussed in the revised **Discussion**.
40. *If branaplam and risdiplam bind the same site then how can ris and ASOi6 be synergistic but not bran and ASOi6?* ► Linear mixture curves (revised **Fig. 6F,G**) show that both the risdiplam/ASOi6 mixture and the branaplam/ASOi6 mixture exhibit synergy. We believe the dose-response curve for branaplam/ASOi6 in the revised **Fig. S13D** does not show significant synergy, because the assessment of synergy via the comparison of three Hill coefficients across three dose-response curves is less sensitive than the assessment of synergy via the observation of a single intermediate maximum in a single linear-mixture curve.
41. *Given that they saw so many null results using the Hill coefficient in this experiment, how can they then say with confidence that risdiplam doesn't bind two sites?* ► The data in **Fig. 5E-H** show that, if risdiplam affects H by binding to the PT, then the resulting effect on splicing is repressive, not activating. This contradicts the two-site hypothesis. It is of course possible that risdiplam physically binds the PT in vivo. But the two-site hypothesis further requires that this binding promote exon inclusion, which we find it does not do. The two-site hypothesis further requires that branaplam not act in the same manner as risdiplam. However, the data in **Fig. 5I-L** shows that, under treatment with branaplam, the value of H is affected by PT deletions in the same way as the value of H observed under treatment with risdiplam. We therefore consider our results to definitively falsify the two-site hypothesis.

42. *All the models analyzed are fully thermodynamic, yet the authors note that there can be kinetic complexities. It seems important to consider what conclusions would be affected by this other class of models. E.g. if the data aren't thermodynamic then can one make conclusions using it about the mode of binding and comparisons to NMR experiments?* ► It is hard to confidently anticipate what conclusions from simpler biophysical models will hold under more complex biophysical models. We still believe it is worthwhile to identify simplified biophysical models that explain data well, even if the original data violate those models in some quantitative respects.

Additional Points:

43. *The reproducibility of biological replicates for RNA-seq experiments and relevant statistics (linear fit parameters, r values, RMSE, etc.) must be reported.* ► In **SI Sec. 1.6** we now report the average Pearson correlation coefficient, $r = 0.988 \pm 0.001$ for the estimated PSI between pairs of samples within the same treatment group; the uncertainty indicates standard deviation across pairs.
44. *Where p values are reported, the authors must specify the statistical test or method for calculating them used.* ► Descriptions of the null hypothesis and computational method used for all p -values have been added; see captions for **Fig. 6** and **Fig. S13**.
45. *The authors should briefly describe methods for how their MPSA libraries were cloned and validated, provide coverage statistics for their pooled plasmid libraries, and describe how they determined if the barcode had an effect on the 5'ss of interest.* ► MPSA library cloning methods have been added to **SI Sec. 1.3**. Coverage statistics have been added to **Fig. S1**: **Fig. S1G** shows the number of barcodes associated with each 5'ss ($\sim 10^3$ to 10^4 for each independently-cloned sub-library); **Fig. S1H** shows the number of total isoform reads observed for each 5'ss in each MPSA experiment ($\sim 10^2$ to 10^4 for each replicate of each sub-library in each treatment condition). The large number of barcodes per 5'ss (**Fig. S1G**) are expected to average out any effects that barcode identity has on 5'ss-specific PSI measurements. The consistency of PSI values observed across sub-libraries with different 5'ss/barcode associations (**Fig. S1A-F**) confirms that barcodes indeed have little effect on 5'ss-specific PSI measurements.
46. *Values for all fit parameters and their errors should be reported (values are not reported for E_{max} , E_{2x} , weights Wx , or for S).* ► The inferred values of all dose-response-curve parameters (S , E_{max} , EC_{2x} , and H) are now reported in **Table S1** for all dose-response curves in **Fig. 5**, **Fig. 6**, and **Fig. S13**. Specifically, this table reports median values and 95% credible intervals, as well as p -values where appropriate. The Hamiltonian Monte Carlo samples from which these values were computed are provided in the GitHub repository.
47. *The multi-drug effects were suggested to exhibit synergy based on 2-3 fold effects. Given the complexity of the system and ad hoc nature of the analysis (especially the variable Hill coefficient) one might want larger effects to be convinced or to interpret these results.* ► We respectfully disagree with this assessment. Synergy was observed for all combinations of drugs tested (except for risdiplam/branaplam, as expected) using linear-mixture curves (**Fig. 6**), and confirmed in over half of the cases using a comparison of Hill coefficients inferred from dose-response curves (**Fig. S13**). This result is important for understanding the mechanisms of splice-modifying drugs. Moreover, effects of 2-3 fold are not small in the context of splice-modifying drugs; such factors can have (and indeed have had) a big impact in the clinic.
48. *FIGURE 2E: It is unclear to me how you can fit the plots to the far right with any confidence in your model. I also ask that the authors comment on the outliers they observe, especially on the far left pair of plots. What are the physical causes of drug insensitivity? Do these represent sites that cannot be bound by the drug for some reason or areas where U1 binding is not rate-limiting for splicing, perhaps? Do they share any common sequence features?* ► Our Bayesian analysis quantifies the precision with which we can infer these curves, and the precision of the resulting drug effect values E . It is true that we observe outliers that deviate significantly from the inferred allelic manifolds, and we note in the revised **Main Text** that this suggests that sequence context outside the 10 bp of the 5'ss can, at least in some cases, impact drug effect.

49. *FIGURE 3: It is not clear in G and H what $\Delta \log W_x$ is in reference to. In the SI, it is explained that it is in reference to the mean for that position. This is difficult to find and difficult to interpret. I would suggest instead picking from the min motifs a consensus base at each position and using that as the referent for $\Delta \log W_x$. By doing this, it becomes clear how mutations lead to free energy penalties and bonuses that can be readily added in an additive model to determine the full effect and what the effect size of each sequence variant is.* ► We have revised the presentation in **Fig. 3** to explicitly discuss inferred values for Gibbs free energy rather than weights. To this end, we have revised the motifs in **Fig. 3** to display $-\Delta\Delta G$ values (in units of kcal/mol) instead of $\Delta \log W$ values. This should make the physical interpretation of these motifs clearer to the reader. However, we believe that logos in the zero-sum gauge (i.e., where the $\Delta\Delta G$ values at each position sum to zero) are easier to understand than logos in the wild-type gauge (i.e., where $\Delta\Delta G$ values are computed relative to a reference sequence). Note that both visualization strategies show the same information. If readers wish to visualize $\Delta\Delta G$ values in different ways, they can do so by modifying and re-running the figure-generation script for **Fig. 3C-H**, which is available in the GitHub repository.
50. *The splicing inhibitors are added for 48 hours when the library is transfected. During that time, the off-target effects of the compounds may cause significant changes in the biology of the system, as seen with the changes to splicing of cell cycle, metabolic, and RNA metabolism transcripts in a recent study that performed 24 hour treatments (Ottesen et al., 2023, NAR). It seems important to do control experiments, for example for different incubation times for a subset of constructs to confirm that the cellular response to the drug is not a confounding factor in the results presented here.* ► We agree with the referee that it is important to assess whether the drugs we study affect global spliceosome activity, as opposed to splicing at specific drug-targeted loci. We have therefore carried out additional control experiments: we assayed the effects of risdiplam, branaplam, ASOi6, and ASOi7 on the *ELP1* minigene (which none of these drugs target), and assayed the effects of RECTAS and ASOi20 on the *SMN2* minigene (which neither of these drugs targets). In all these experiments we see no substantial off-target drug effect (**Fig. S12**). We believe these data address the referee's primary concern, and that additional studies involving time-course experiments, though they could be illuminating, are not necessary to support our conclusions.
51. *Are there DMSO effects? The authors appropriately compare results of drugs added in DMSO to DMSO alone, but it would be of interest to know if there is an effect of this amount of DMSO alone.* ► Previous literature has reported effects of DMSO on splicing, e.g. (Bolduc et al., 2001). However, all of our experiments were carried out in matching DMSO conditions, so any effects that DMSO might have are controlled for.
52. *Also, as noted above, the drug is present for an extended period of time; is there evidence for or against significant metabolism of the drug over this time?* ► Our treatment protocols are similar to those in relevant prior work (Ajiro et al., 2021; Naryshkin et al., 2014; Sivaramakrishnan et al., 2017; Wang et al., 2018). In vivo pharmacokinetic studies for risdiplam (Ratni et al., 2021) have found a half-life of 6.4 hr in rats and 5.4 hr in cynomolgus monkeys. In vivo pharmacokinetic studies for branaplam tool compounds have found half-lives of 3-5 hr in mice (Cheung et al., 2018). However, it is unclear whether these half-lives are relevant in the HeLa cell cultures that we used. While it would be nice to have more information about how these drugs are metabolized, we believe that the standardized culture conditions that we used throughout our studies make a deeper understanding of drug metabolism unnecessary for the claims of the current manuscript.
53. *What fraction of data was thrown out for "alternative splicing events" in each sample? Are there trends observed and are these sequences available for others to analyze?* ► As now stated in the revised **SI (Sec. 1.6)**, we focused our RNA-seq analysis on cassette exon events. Specifically, we selected 247,321 cassette exon events (~68% of the 365,260 events reported by rMATS) having a median total counts of at least 10 across samples and at least a single read supporting exon skipping in any of the samples. We then used a Bayesian multiple logistic regression model to simultaneously infer PSI values and corresponding 95% posterior credible intervals for all three drug treatments for 235,711 distinct exons having 5'ss sequences with GT at +1 and +2. Among these exons were 13,431 distinct 10 nt 5'ss sequences. Other alternative splicing events (including cassette exons with non-GT 5'ss, alternative 5'ss usage, alternative 3'ss usage, and intron retention) were not analyzed. Our RNA-seq data is publicly available (in NCBI GEO), as is the output of our rMATS

analysis (on GitHub). Our MPSA data did not distinguish between alternative splicing events other than exon inclusion/exclusion.

54. *Figures 1D,E,H,I all have a point that is differently colored at graph coordinates (10,100). What is the significance of this?* ► Comment now pertains to **Fig. 1D,E**. The black dot indicates the wild-type *SMN2* exon 7 5'ss (AGGA/GUAAGU). The **Fig. 1** caption has been revised to clarify this.
55. *Figure S2A-E. An analogous graph with all sequences with consensus base (for S2A, -4A as an example) would be helpful.* ► Comment now pertains to Fig. S8. This figure is only intended to show that single nucleotide deviations from the most permissive risdiplam IUPAC motif do in fact abrogate activation by risdiplam. **Fig. S3B** shows that all 5'ss that match the permissive risdiplam IUPAC motif do in fact satisfy the classification criteria; this fact has been clarified in the **Fig. S8** caption.
56. *I ask the authors to clarify the method of Bayesian parameter fitting. Is this a maximum a posteriori estimate, from MCMC sampling, etc.?* ► Bayesian models for allelic manifolds describing drug effect (**Fig. 2**) were defined and analyzed in STAN. Bayesian models for 5'ss-specific drug effect [the two-interaction-mode model (**Fig. 3**) and one-interaction-mode model (**Fig. S6**)] were defined and analyzed in numpyro. Bayesian models for concentration-dependent drug effect (**Fig. 5**) were defined in numpyro. For all Bayesian models, posterior parameter values were sampled using the No-U-Turn Sampler, which is a type of Hamiltonian Monte Carlo sampler. Reported parameter values reflect medians and 95% credible intervals of these posterior samples. These facts have been clarified in **SI Sec. 4**.
57. *“Allelic manifold” seems like a needlessly complicated and potentially misleading term. A manifold is an abstract topological space with locally Euclidean behavior, where “locally Euclidean” is defined by the preservation of certain continuity, differentiability, or geometric properties under invertible maps from open subsets of the manifold to subsets of Euclidean space (Lee, 2012, Intro. to Smooth Manifolds, Springer GTM). While the graph of any smooth function is a smooth manifold, the manuscript neither needs nor uses any of the mathematical or conceptual technology of manifold theory to do its comparatively-simple statistical mechanical analyses, nor is there any focus on specific allelic differences/variants. I would suggest instead referring to this as a “statistical mechanical model” or “phenomenological model”. (I note that a previously published paper used the “allelic manifold” terminology [Forcier et al., 2018, eLife]; in that paper’s published reviews, some reviewers also found this term to be confusing.)* ► We respectfully disagree with this assessment. We believe the terminology “allelic manifold” is both accurate and helpful, and we have revised the manuscript (**Main Text** and **SI Sec. 3.1**) to better describe why this is. Specifically, this allelic manifold formulation is useful because it reveals that, if the biophysical model is correct, then measurements plotted in the three-dimensional space ($\Psi_{\text{DMSO}}, \Psi_{\text{ris}}, \Psi_{\text{bran}}$) should collapse to a one-dimensional curve. This one-dimensional curve in 3D space satisfies the mathematical definition of a manifold. The shape of the allelic manifold is determined by the drug effect values E_{ris} and E_{bran} , whereas locations within the allelic manifold are determined by context strength S . The term “allelic” refers to the different sequence contexts acting as an allelic series. The coordinates ($\Psi_{\text{DMSO}}, \Psi_{\text{ris}}, \Psi_{\text{bran}}$) act as different phenotypes for alleles in that series. Note also that this concept of an allelic manifold is not the same as a statistical mechanical model. The statistical mechanical model (the drug-effect model in **Fig. 2**) specifies the dimensionality of the manifold as well as its shape in the embedding space, but the general phenomenon of data collapse inherent to the allelic manifold concept is agnostic to the specific form of the underlying model considered.
58. *An interactive process is used to redefine the hyp motif based on MPSA and genome-wide data—and the authors acknowledge the models are still qualitative at this point.* ► We have revised the **Main Text** to explicitly describe the IUPAC motif analysis as qualitative and exploratory, as well as to clarify its rationale.
59. *Line 292: “bran motif” should be changed to “hyp motif”.* ► This error is fixed in the revised text.
60. *The logo motif vs. NMR structure comparison may not be as straightforward or conclusive as suggested by the authors. The NMR study doesn’t have the same bases but it still binds the drug.* ► The revised text clarifies our main point—that our results for risdiplam’s 5'ss specificity are partially consistent with the NMR structure, but that the NMR structure does not obviously explain all aspects of

risdiplam's 5'ss specificity. In particular, the NMR structure does not explain why risdiplam is specific for -4A in the 5'ss, and this is precisely because there is no position -4 in the 5'ss RNA present in the NMR structure.

61. *If each tautomeric form of branaplam binds differently, then it would be (highly) unlikely that they are energetically equivalent.* ► We do not claim that the different tautomers of branaplam bind equivalently. We only say that the different tautomeric forms of branaplam provide one potential explanation for the two-interaction-mode hypothesis.
62. *The MD simulation (shown for one tautomer) seems superfluous. If there are more rotatable bonds there's more conformational entropy and chances to bind in different conformations.* ► The MD simulations are provided only to quantitatively confirm the qualitative expectation (from the planar structures in **Fig. 4D,E**) that the additional rotatable bonds in branaplam relative to risdiplam do indeed introduce additional conformational entropy. We present MD simulations of the enol tautomer of branaplam, because this is the tautomer of branaplam with the most rotatable bonds. The results confirm that branaplam does indeed have more flexibility than risdiplam.
63. *Why use EC_{2x} instead of EC₅₀? It is more difficult to interpret EC_{2x}.* ► We have clarified this point in the **Main Text** and **SI Sec. 3.4**. Specifically, (1) the equation for the Gibbs free energies of the biophysical model has a simple expression in terms of EC_{2x} but not EC₅₀; (2) EC_{2x}, but not EC₅₀, can be confidently determined from dose-response data even in the absence of saturation; and (3) the value of EC_{2x} is not impacted by the value of ΔG_{U1} , whereas the value for EC₅₀ is. Point (2) is particularly important because many of the drugs whose dose-response curves we measure exhibit lethality before the concentration at which their splice-modifying effect saturates.
64. *In the SI, the authors mention that they chose the drug concentrations from pilot dose-response experiments where they measured EC_{2x}. They should show the data from these experiments in a supplemental figure and define what the EC_{2x} is, how it was determined, and in what context it was measured, as well as provide errors on these parameters.* ► The pilot dose-response data and inferred curves are now shown in **Fig. S1J,K** and described in **SI Sec. 1.6**.
65. *When concentrations 7.1x and 10x above the EC_{2x} were chosen for each drug, is this within the plateau of the dose-response regime? As a control, I suggest that the authors also perform experiments using other drug concentrations to rule out differences arising from different treatment concentrations.* ► We thank the reviewer for raising this subtle but important point. The answer is no, the drug concentrations of 7.1x (used for the MPSA experiments) and 10x (used for RNA-seq experiments) are not in the saturating regime for either risdiplam or branaplam. This concentration difference is equivalent to a difference of $RT \log(10/7.1) = 0.211$ kcal/mol between the chemical potential of the drugs in the RNA-seq experiment vs. in the MPSA. To compensate for this difference, a 0.211 kcal/mol shift in chemical potential is applied when computing the predictions of the two biophysical models for 5'ss specificity (both the two-interaction-mode model and one-interaction-mode model) on RNA-seq data relative to MPSA data. This chemical potential shift is now explicitly described in the revised **SI, Sec. 3.2**.
66. *Line 484: Kinetic proofreading requires net energy input (usually ATP hydrolysis; Ninio, 1975, Biochemie; Hopfield, 1974, PNAS). Where would that come from?* ► Multiple DExD/H-box ATPase components of the spliceosome are known to play important (ATP-dependent) roles in ensuring spliceosome fidelity (Semlow and Staley, 2012). We now mention this in the revised **Discussion**.
67. *I follow the logic of validating known targets of these drugs, but with a genome wide screen why not go a step further to describe off targets, new targets, etc.?* ► Some known off-targets of risdiplam and branaplam are highlighted in **Fig. 2D** and discussed in the **Main Text**. **Fig. 2D** confirms that our RNA-seq analysis results are consistent with existing knowledge for off-targets of these drugs. It would indeed be possible for us to describe additional splicing events in the human transcriptome that are promoted by risdiplam and branaplam. We are reluctant to do this, however, because it is unclear how such an analysis would further the primary claims of our paper, which focuses on locus-specific drug mechanisms. We also note that other studies have already provided discussions of the off-target effects of risdiplam and branaplam (Keller et al., 2022; Krach et al., 2022; Monteys et al., 2021; Ratni et al., 2018; Sivaramakrishnan et al., 2017).

68. *Along those lines, we suggest that the authors comment on and compare their work to the recent similar study by Ottesen et al. (2023) in Nucleic Acids Research. In that study, the authors used RNA-seq to measure the off-target effects of branaplam and risdiplam transcriptome-wide and with mini-gene reporters. Does the model in this study predict those effects and their sizes? How do transcriptome-wide data from this study compare?* ► We believe it is beyond the scope of this paper to re-analyze the data from Ottesen et al. (Ottesen et al., 2023), which was published after our original manuscript submission. Ottesen et al. is not the first paper to analyze the off-target effects of risdiplam and branaplam transcriptome-wide using RNA-seq. **Fig. 2D** and its accompanying discussion already provide a comparison of our on-target and off-target predictions based on the findings of multiple other groups (Keller et al., 2022; Krach et al., 2022; Monteys et al., 2021; Ratni et al., 2018; Sivaramakrishnan et al., 2017). We note this in the revised **Main Text** and have added citations to Ottesen et al. where appropriate. Regarding the minigene data, Ottesen et al. reports PSI measured by RT-PCR with ethidium bromide staining (with only one replicate). Because of this we do not believe we can productively reanalyze these data using our quantitative models.

Response to Reviewer #3

#3 (Remarks to the Author):

Ishigami, Wong et al report a biophysical model to quantitatively define the titration effects and sequence specificity of the 5' splice site-modifying drugs risdiplam and branaplam, which are currently used (risdiplam) or proposed to be used (branaplam) as therapies for Spinal Muscular Atrophy, and in the case of branaplam also as a potential therapy for Parkinson's disease. Their results provide molecular models to explain the distinct sequence specificities of these drugs, arguing that branaplam (but not risdiplam -contrary to previous proposals-) uses two distinct interaction modes to recognize 5' splice sites. In addition, the authors report cooperativity and synergy effects between these drugs and splice-switching antisense oligonucleotide drugs.

Quantitative modeling of the effects of splicing-modifying drugs will be key to develop rigorous pharmacological understanding of the mechanisms of action of these compounds and rationally design drugs of improved activity and specificity. The work by Ishigami, Wong et al provides a very valuable framework to approach these goals. Among other insights, the results reveal sequence specificity determinants for drug specificity that go beyond the "bulge repair" mechanism inferred from high resolution structural data, a popular model whose appealing simplicity is challenged by the new results of Ishigami, Wong et al. The paper reports an important volume of work and an original combination of high-throughput mutagenesis and transcriptome data, biophysical modeling and careful drug titration experiments.

► We thank the referee for this favorable assessment.

In my opinion the following revisions would help to improve the manuscript:

1. *A general comment is that at various points the manuscript is challenging to follow and therefore would benefit from additional explanations and clarity.* ► We agree that the initial manuscript was unnecessarily confusing. We have made substantial changes to improve clarity throughout the **Main Text** and **SI**.
 - a. *For example, the text concerning panels 1D-1I soon becomes a bit cryptic and would benefit from additional explanations of the exact meaning of "ris min", "ris max", "hyp min" and "hyp max".* ► We have greatly simplified the discussion of the IUPAC motifs. **Fig. 1D-F** and **Main Text** now discusses only a "risdiplam IUPAC motif" and a "hyper-activation IUPAC motif". Discussions of the former "min" and "max" motifs—now called the "restrictive" and "permissive" motifs—is confined to the **SI (Sec. 2.3 and Fig. S3)**, and is presented only to convey the range of IUPAC motifs consistent with the MPSA data.
 - b. *Similarly, the text can become cryptic for non-specialists when describing the validation of the biophysical modelling efforts or the falsification of the two-site hypothesis for risdiplam. The final messages are clear, but the narrative leading to them could be streamlined.* ► These concerns are addressed by the revised manuscript. The revised text now clearly distinguishes between the exploratory analysis that yields the IUPAC motifs (**Figs. 1,2**), the biophysical modeling analysis of sequence-dependent drug activity (**Fig. 3**), and the quantitative empirical modeling

of titration curves (**Figs. 5,6**). The biophysical modeling section also now uses more physically meaningful terminology (e.g., “energy” instead of “weight”). Finally, the section describing falsification of the two-site hypothesis for risdiplam has been revised for clarity.

2. Figure 1D/E:

- a. *A substantial number of “ris min match” and -specially- “ris max match” are located in the diagonal and therefore they show not much difference in PSI values between DMSO and risdiplam, i.e. a number of 5' ss matching the motifs are not really affected by the drug, arguing that the presence of the motif is insufficient to confer sensitivity to the drug at low or high PSI values. This is easier to explain at high PSI values, as there is little room to see enhanced exon inclusion, but more difficult to explain at low PSI values, where there is ample margin for detection of improved inclusion levels.* ► As described in the revised **Main Text**, sensitivity at low PSI is limited by background due to the use of cryptic splice sites, i.e., 5'ss at other locations in the minigene. In our experience, a low level of cryptic splice site usage is common in MPSA experiments due to the high sensitivity of the assay. In the *SMN2* minigene, we did remove the most active cryptic 5'ss prior to performing the MPSAs (**SI Sec. 1.3**), but lower activity cryptic 5'ss remained. The presence of splicing at cryptic splice sites is reflected in the nonzero splicing of the negative control 5'ss at PSI of ~0.3 (**Fig. S1**). It is therefore not surprising that some 5'ss that match the motif but have basal PSI near or below 0.3 do not appear to be responsive to drug (i.e., also have treatment PSI near or below 0.3), since basal PSI near or below 0.3 is most likely due to splicing at a cryptic 5'ss, not the motif-matching 5'ss.
- b. *Related to this latter point: a substantial number of measurements appear to fall in the range between 1% and 0.1% exon inclusion, are these differences reliable? (see also similar points below)* ► Measured differences near PSI of ~0.3 are indeed reliable, as indicated by the variation between biological replicates (**Fig. S1**). The interpretation of these differences is a different question; we argue that PSI values down to ~0.3 can be interpreted as being specific to the variable 5'ss, but that PSI values below ~0.3 are likely primarily due to the use of cryptic 5'ss.
- c. *The presentation of the data on the effects of branaplam against DMSO is confusing, because they are shown (albeit with a different format) in Figure S3 (by the way, Figure S3 is labeled as Figure S5 in the figure itself [This has been corrected.]) and again in Figures 1H and 1I, but without an associated motif. The Figure may be easier to follow if panels H and I are swapped with F and G, such that the effects of individual drugs are seen first, and then the comparison of the differential effects between the two drugs.* ► We believe our revisions will address the referee's concerns about clarity. The revised **Fig. 1D** shows risdiplam vs. DMSO and highlights matches to the risdiplam IUPAC motif; the revised **Fig. 1E** shows branaplam vs. risdiplam and highlights matches to the hyper-activation IUPAC motif; and the revised **Fig. 1F** shows branaplam vs. DMSO and highlights matches to both the risdiplam IUPAC motif and the hyper-activation IUPAC motif. This order is chosen so that each motif is first presented individually, and then the effects of the two motifs are combined. A similar format is used in the revised **Fig. S3**. The revised **Fig. S4** (previously Fig. S3) illustrates the impossibility of describing observed branaplam vs. DMSO effects using a single IUPAC motif.
- d. *It is not straightforward to conclude, from Figure S3 alone, “that the minimally permissive IUPAC motif that matches all 5'ss in class 1 also matches some 5'ss in class 2”;* ► We have clarified in the caption to the revised **Fig. S4** (formerly Fig. S3) that the dots highlighted are the prediction of the most restrictive motif that matches all activated 5'ss (those in region 1b), and that the fact that this most restrictive motif also matches 5'ss that are not sensitive to branaplam (those in region 2b) means that there are not any IUPAC motifs that match all activated 5'ss and no insensitive 5'ss.
- e. *I am not convinced, at this point in the paper, that this conclusion is granted: “These observations are consistent with a “two-motif” model for branaplam, i.e., one in which branaplam recognizes two distinct but overlapping classes of 5'ss: 5'ss also activated by risdiplam, and 5'ss hyper-activated by branaplam relative to risdiplam.” Why should the results reported at this point of the paper necessarily invoke a “two-motif” model, rather than simply lower sequence*

specificity requirements for branaplam? ► IUPAC motifs are capable of describing a wide range of specificities, ranging from exact sequence matches to ambiguous matches at many or all nucleotide positions. This analysis, which we have clarified in the revised **Main Text**, shows that no single IUPAC motif can explain the effects of branaplam (shown in **Fig. S4**), but that a two-IUPAC motif model can explain the effects of branaplam (shown in **Fig. 1F**), therefore suggesting a two-interaction-mode model for branaplam. Indeed, it was this observation in the behavior of IUPAC motifs on MPSA data that originally led us to propose the biophysical model, described in **Fig. 3**, that formalizes the two-interaction-mode hypothesis. That being said, the revised **Main Text** emphasizes that the IUPAC analysis in **Fig. 1** and **Fig. 2** was exploratory, and that the formal assessment of the two-interaction-mode hypothesis requires the comparison of explicit biophysical models (**Fig. 3**, **Fig. S6**, and **Fig. S7**).

- Figure 2E: “We note, however, that some data points deviate significantly from their corresponding inferred allelic manifolds; this indicates that genetic context outside of the 10-nt 5’ss sequence has a detectable (if secondary) influence on drug effect. Understanding the mechanistic basis for this secondary influence on drug effect is a worthy goal for future research.” The two examples that differ significantly from inferred allelic manifolds (SMN2 and FOXM1) are those in the diagonal of panel 4D [we believe this refers to Fig. 2E], while the two that fit nicely the curves (SF3B3 and HTT) show much higher E values for branaplam than risdiplam. Does this say anything about the model being far more accurate to predict sequence determinants that determine hyperactivation by branaplam, but less so for common sensitivity to risdiplam / branaplam? Can this be used to further refine the model?* ► This is a fair question, but we suspect the referee’s observation results primarily from the fact that different 5’ss correspond to different numbers of exons with intermediate PSI. The key issue is that exons with intermediate PSI under both DMSO and drug treatments have more opportunity to visually deviate from the inferred allelic manifold than do exons that have high or low PSI under either DMSO or drug treatment. For the SMN2 5’ss and FOXM1 5’ss, there are a few dozen exons that have intermediate PSI under both DMSO and drug treatment, and ~10% of these appear to deviate substantially from the inferred manifold. For the HTT 5’ss and the SF3B3 5’ss, on the other hand, there are very few exons that have intermediate PSI under both DMSO and drug treatments, and therefore it is difficult to tell whether ~10% of these exons deviate significantly from the inferred manifold. Also, **Fig. 2E** shows only a small set of the inferred allelic manifolds. A systematic analysis of deviations from the inferred allelic manifolds for all 5’ss would be needed to determine if the referee’s observations hold generally. This would be an interesting analysis to do, but it is nontrivial and, we feel, better suited to future work.
- Figure 3: the authors conclude that “Plotting these data against model predictions confirms that the two interaction-mode model for branaplam explains these data for both drugs well (Fig. 3C-F) and better than the one-interaction-mode model for branaplam does (Fig. S12).” R2 values for the two-interaction mode in panels C-F are really not so different from those of the one-interaction mode shown in S12: 0.400 vs 0.403, 0.986 vs 0.986, 0.634 vs 0.601 and 0.960 vs 0.939. Based on these data alone, the two-interaction mode seems to be only marginally statistically better than the one-interaction model.* ► The change in how well the two-interaction-mode model vs. one-interaction-mode model explains activation by branaplam (R^2 of 0.634 vs 0.602 for RNA-seq data, R^2 of 0.958 vs 0.936 for MPSA data) is highly statistically significant, though quantitative and somewhat subtle. The statistical power of this difference is best quantified by the log likelihood ratio, which (unlike R^2) accounts for the quantitative properties of experimental noise. Our results show a highly statistically significant log likelihood ratio of 117.4, with 95% credible interval of [64.4, 172.5]. We have revised the **Main Text** to clarify this point.
- The statement “One prediction of the two-site hypothesis for risdiplam is that a molecule of risdiplam bound to the PT should promote SMN2 exon 7 inclusion even when a second molecule of risdiplam is prevented from binding to the U1/5’ss complex.” may not be necessarily reflect the prediction of the two-site hypothesis (as far as I understand it): the two-site hypothesis may require concerted actions from the two sites to effectively induce exon skipping, one hit not being sufficient for achieving detectable effects.* ► This is a fair critique. In the revised text, only the Hill coefficients observed for the dose-response curves in **Fig. 5E-H** are used to argue against the two-site hypothesis. Our observation that deleting the PT increases the Hill coefficient, rather than reducing it, provides what we believe is definitive evidence against all formulations of the two-site hypothesis, including a formulation in which

the effect of risdiplam at the PT is contingent on a molecule of risdiplam being bound to the U1/5'ss complex.

6. *Figure 5: it would be good to discuss how the determination of Hill coefficients may be affected by the intrinsic inaccuracy of measurements of inclusion/skipping ratios, in particular with ratios above 10 or below 0.1 (e.g. a significant number of measurements are between 0.1 and 0.01 or even below: are measurements below 5% of exon inclusion accurate, reproducible and trustable)?* ► The effect of noise in the inclusion/exclusion ratio measurements on the inferred Hill coefficient is explicitly accounted for in our Bayesian modeling procedure. Moreover, while the precision of measured inclusion/exclusion ratios at low (and at high) PSI would be a major concern had we measured these ratios using radioactive gels, our use of RT-qPCR instead allowed us to measure these ratios at high precision over many orders of magnitude, including PSI values substantially below 5 and substantially above 95. The precision in our measurements is apparent in the small amount of scatter between the 2 or 3 biological replicates measured at each drug concentration in **Figs. 5, 6, S12, and S13**; note that the dots indicating the 2 or 3 biological replicates at each drug concentration often overlap.
7. *In panels 5I-L branaplam is wrongly spelled (branplam).* ► We thank the referee for noticing these typos; they have been fixed.
8. *Figure 6: as for the previous point, a similar question of accuracy in the determination of inclusion values above 10 or below 0.1 could be made. This is particularly relevant for panels L to O, in which the measurements range from 90-91% inclusion to 93-96% inclusion.* ► Our above response to critique 6 (concerning **Fig. 5**) also addresses this similar concern about **Fig. 6**.
9. *The synergy effects of Figure 6 may not be easy to follow because the effects of risdiplam / branaplam are not in the same figure, so one needs to jump between figures to compare Hill coefficients and it is not clear what comparisons correspond to what statistical significance values. It would be great to have a Table in which the comparison between Hill coefficients would be straightforward to follow.* ► This is a good suggestion. We have included a new table (**Table S1**) that lists the Hill coefficients and corresponding P-values for the inferred dose-response curves in **Figs. 5, 6, and S13**.
10. *The sentence “These findings definitively establish that a splice-modifying drug can exhibit cooperativity even when it binds only a single site on target pre-mRNA, and that the extent of this cooperativity can differ between drugs that promote inclusion of the same cassette exon.” assumes that drugs bind to a single site on target pre-mRNA, but this is not necessarily the case, particularly when using ASOs that may have a single target with 100% complementarity, but that may engage with other sites, even if not fully complementary.* ► We have softened this statement to “We conclude that a splice-modifying drug can exhibit cooperativity even when it binds only a single site on target pre-mRNA, and that the extent of this cooperativity can differ between drugs that promote inclusion of the same cassette exon.” We believe this statement is supported by the ASOi7 data, especially in light of new dose-response data (**Fig. S12D**), which shows ASOi7 (which exhibits marked anomalous cooperativity on the *SMN2* minigene) has no detected effect on splicing of the *ELP1* minigene, thereby establishing that ASOi7 does not have widespread off-target effects on splicing in general. We also note that a BLASTN analysis confirmed that ASOi7 does not have significant complementarity to off-target sites within the *SMN2* minigene pre-mRNA.

Response to Reviewer #4

Reviewer #4 (Remarks to the Author):

The problem is certainly relevant, and a solid clarification of the mode of action (MOA) of these 2 drugs would be highly informative and impactful. However, the work is quite complex and hard to follow for someone who is not too familiar with this kind of parallel splicing assays. Reading the paper, I got easily lost. ► We apologize for the lack of clarity and accessibility of the initial manuscript. We believe the revised manuscript is much clearer and more accessible. In particular, the discussion of the IUPAC motifs derived from MPSA data (in **Fig. 1**) has been simplified, and non-critical details have been moved to **SI**. The discussion of quantitative modeling, both the methods and the motivation, have also been clarified.

For example, the first paragraph of the results is more about the methodological approach, rather than 'results'.

► This is a good point. We have removed the first paragraph of the **Results** section. The content of this paragraph is now covered in the second paragraph of the revised **Discussion**.

The overall indication of the MOA is potentially interesting, but all these indirect observations still leave quite some room for potential interpretations that may differ from those reported. ► If the MOA here refers to the two-site hypothesis for risdiplam, we believe the dose-response measurements in **Fig. 5** directly refute this MOA. If the MOA here refers instead to the two-interaction-mode model for branaplam, we agree that the data for this model are indirect. Still, we do obtain support for the two-interaction-mode model using two orthogonal experimental methods (MPSAs and RNA-seq), and the statistics supporting the two-interaction-mode model are highly significant. Finally, the revised **Discussion** now proposes specific structural studies that could directly confirm or refute the two-interaction-mode model.

There are several instances in which drug cooperativity and multi-drugs synergy is used to argue about drug efficacy and MOA. I admit my limitations here, but may the overall story could be made easier to follow and more efficient in describing what would be the ultimate message - is it about a massive experimental framework for characterizing RNA-targeted small molecules or is it about the ultimate clarification of the MOA of risdiplam and branaplam ? ► The revised text clarifies the central point of our paper: that quantitative mechanistic modeling is important when trying to infer the MOA of a splice-modifying drug from functional data. In particular, the revised text better illustrates how quantitative mechanistic modeling can (1) be used to integrate different types of data (e.g., MPSA and RNA-seq data) into a single mechanistic model; and (2) can deconvolve different mechanistic influences (e.g., context strength, drug effect, EC_{2x} , and Hill coefficient) even when those influences cannot be separated experimentally.

Other drugs are mentioned, like RECTAS and BPN-154. What about these? Are not included because this framework is too costly? too much time-consuming? I understand the paper wants to use this experimental framework on those two drugs, but I found it confusing (to me). ► We performed experiments on RECTAS (**Fig. 6C,K**) only to test whether our findings of anomalous cooperativity and multi-drug synergy, observed in the context of drugs that target SMN2 exon 7, also applied to drugs that target other loci. As stated in the revised **Discussion**, it would be interesting to use the methods we describe to dissect the sequence determinants of RECTAS activity, as well as the sequence determinants of the activities of BPN-15477 and kinetin. However, such studies are not needed to support the claims of the present manuscript. Moreover, such studies would require a substantially modified experimental and modeling approach, as RECTAS appears to target the kinase CLK1, not the U1/5'ss complex.

References cited

- Ajiro M, Awaya T, Kim YJ, Iida K, Denawa M, Tanaka N, Kurosawa R, Matsushima S, Shibata S, Sakamoto T, Studer R, Krainer AR, Hagiwara M. 2021. Therapeutic manipulation of IKBKAP mis-splicing with a small molecule to cure familial dysautonomia. *Nat Commun* **12**:4507. doi:10.1038/s41467-021-24705-5
- Bolduc L, Labrecque B, Cordeau M, Blanchette M, Chabot B. 2001. Dimethyl Sulfoxide Affects the Selection of Splice Sites*. *J Biol Chem* **276**:17597–17602. doi:10.1074/jbc.m011769200
- Cheung AK, Hurley B, Kerrigan R, Shu L, Chin DN, Shen Y, O'Brien G, Sung MJ, Hou Y, Axford J, Cody E, Sun R, Fazal A, Fridrich C, Sanchez CC, Tomlinson RC, Jain M, Deng L, Hoffmaster K, Song C, Hoosear MV, Shin Y, Servais R, Towler C, Hild M, Curtis D, Dietrich WF, Hamann LG, Briner K, Chen KS, Kobayashi D, Sivasankaran R, Dales NA. 2018. Discovery of small molecule splicing modulators of survival motor neuron-2 (SMN2) for the treatment of spinal muscular atrophy. *J Med Chem* **61**:11021–11036. doi:10.1021/acs.jmedchem.8b01291
- Keller CG, Shin Y, Monteys AM, Renaud N, Beibel M, Teider N, Peters T, Faller T, St-Cyr S, Knehr J, Roma G, Reyes A, Hild M, Lukashev D, Theil D, Dales N, Cha J-H, Borowsky B, Dolmetsch R, Davidson BL,

- Sivasankaran R. 2022. An orally available, brain penetrant, small molecule lowers huntingtin levels by enhancing pseudoexon inclusion. *Nat Commun* **13**:1150. doi:10.1038/s41467-022-28653-6
- Krach F, Stemick J, Boerstler T, Weiss A, Lingos I, Reischl S, Meixner H, Ploetz S, Farrell M, Hehr U, Kohl Z, Winner B, Winkler J. 2022. An alternative splicing modulator decreases mutant HTT and improves the molecular fingerprint in Huntington's disease patient neurons. *Nat Commun* **13**:6797. doi:10.1038/s41467-022-34419-x
- Monteys AM, Hundley AA, Ranum PT, Tecedor L, Muehlmann A, Lim E, Lukashev D, Sivasankaran R, Davidson BL. 2021. Regulated control of gene therapies by drug-induced splicing. *Nature* **596**:291–295. doi:10.1038/s41586-021-03770-2
- Naryshkin NA, Weetall M, Dakka A, Narasimhan J, Zhao X, Feng Z, Ling KKY, Karp GM, Qi H, Woll MG, Chen G, Zhang N, Gabbeta V, Vazirani P, Bhattacharyya A, Furia B, Risher N, Sheedy J, Kong R, Ma J, Turpoff A, Lee C-S, Zhang X, Moon Y-C, Trifillis P, Welch EM, Colacino JM, Babiak J, Almstead NG, Peltz SW, Eng LA, Chen KS, Mull JL, Lynes MS, Rubin LL, Fontoura P, Santarelli L, Haehnke D, McCarthy KD, Schmucki R, Ebeling M, Sivaramakrishnan M, Ko C-P, Paushkin SV, Ratni H, Gerlach I, Ghosh A, Metzger F. 2014. SMN2 splicing modifiers improve motor function and longevity in mice with spinal muscular atrophy. *Science* **345**:688–693. doi:10.1126/science.1250127
- Ottesen EW, Singh NN, Luo D, Kaas B, Gillette BJ, Seo J, Jorgensen HJ, Singh RN. 2023. Diverse targets of SMN2 -directed splicing-modulating small molecule therapeutics for spinal muscular atrophy. *Nucleic Acids Res.* doi:10.1093/nar/gkad259
- Ratni H, Ebeling M, Baird J, Bendels S, Bylund J, Chen KS, Denk N, Feng Z, Green L, Guerard M, Jablonski P, Jacobsen B, Khwaja O, Kletzl H, Ko C-P, Kustermann S, Marquet A, Metzger F, Mueller B, Naryshkin NA, Paushkin SV, Pinard E, Poirier A, Reutlinger M, Weetall M, Zeller A, Zhao X, Mueller L. 2018. Discovery of risdiplam, a selective survival of motor neuron-2 (SMN2) gene splicing modifier for the treatment of spinal muscular atrophy (SMA). *J Med Chem* **61**:6501–6517. doi:10.1021/acs.jmedchem.8b00741
- Ratni H, Scalco RS, Stephan AH. 2021. Risdiplam, the first approved small molecule splicing modifier drug as a blueprint for future transformative medicines. *ACS Med Chem Lett* **12**:874–877. doi:10.1021/acsmchemlett.0c00659
- Semlow DR, Staley JP. 2012. Staying on message: ensuring fidelity in pre-mRNA splicing. *Trends Biochem Sci* **37**:263–273. doi:10.1016/j.tibs.2012.04.001
- Sivaramakrishnan M, McCarthy KD, Campagne S, Huber S, Meier S, Augustin A, Heckel T, Meistermann H, Hug MN, Birrer P, Moursy A, Khawaja S, Schmucki R, Berntenis N, Giroud N, Golling S, Tzouros M, Banfai B, Duran-Pacheco G, Lamerz J, Liu YH, Luebbbers T, Ratni H, Ebeling M, Cléry A, Paushkin S, Krainer AR, Allain FH-T, Metzger F. 2017. Binding to SMN2 pre-mRNA-protein complex elicits specificity for small molecule splicing modifiers. *Nat Commun* **8**:1476. doi:10.1038/s41467-017-01559-4
- Wang J, Schultz PG, Johnson KA. 2018. Mechanistic studies of a small-molecule modulator of SMN2 splicing. *Proc Natl Acad Sci USA* **115**:201800260. doi:10.1073/pnas.1800260115

REVIEWERS' COMMENTS

Reviewer #2 (Remarks to the Author):

Ishigami et al. have addressed the core concerns of the reviewers. By putting their analysis on a firmer biochemical footing, they have shown a proof-of-principle of their method to quantitatively interrogate drug specificity and have greatly enhanced their paper and its ability to dissect the binding of splice site targeting drugs. The methods and supplemental text present an excellent resource for anyone trying to apply similar conceptual and experimental approaches to other biological systems. In general, the extra precision, clarity, and detail with the revision are much appreciated. With that said, I have a few minor points for revision or clarification:

1. The biophysical models are completely thermodynamic under the assumption that U1 binding is strongly rate limiting for splicing. However, you argue from the empirical models that kinetic models may be required to explain splicing outcomes. I think the paper would benefit from a very brief consideration of how conclusions would or would not be altered in kinetic models where drugs can alter splicing rates, not just U1 occupancy.

2. Importantly, such considerations probably obviate the statements that the two-site model is "disproven". It is very hard to disprove all models or know when all have been considered, and given the uncertainty in the biophysical models (Hill slopes $\neq 1$) I would caution against statements this strong.

3. That said, the manuscript argues that their data argue against binding by two drug molecules. What would the dose-response curves look like with a Hill coefficient constrained to 2. How much worse is this at explaining the data, and what are the log-likelihoods of having integer vs. floating Hill coefficients? This analysis would help strengthen the conclusion against multiple binding events.

4. Along the same lines, in lines 309-312, the authors hypothesize that in a dsRNA context, two molecules of drug could bind in flipped configurations, which would seem to contradict the conclusion that multiple binding does not occur. Could you please clarify the implications of such a model and whether it is consistent with your data?

5. The discussion of synergy could still be clarified.

a. The choice of a linear mixture model was not immediately intuitive to this reviewer. The model proposed for synergy amounts to independent binding to multiple sites by multiple drugs. The total effect is then modeled as a linear combination of the distinct effects. However, it is not immediately clear how the molecular model connects to the parameters of the linear mixture model.

b. The description of the different expectations for having synergy vs. no synergy could be made clearer by showing mock data plots or simulations for each case.

c. More simply, different authors mean different things by "Synergy" so the authors should carefully define how they are using this term and define it physically and mathematically even if not simulated.

Reviewer #3 (Remarks to the Author):

The authors have addressed the issues raised in my previous report and I recommend publication of the manuscript in Nature Communications.

Juan Valcarcel

Response to reviewer comments

In what follows, referee comments are marked in *black italic text*, while our point-by-point responses to the referee comments are marked in **blue regular text** and preceded by a ► character.

Response to reviewer #2

Reviewer #2 (Remarks to the Author):

Ishigami et al. have addressed the core concerns of the reviewers. By putting their analysis on a firmer biochemical footing, they have shown a proof-of-principle of their method to quantitatively interrogate drug specificity and have greatly enhanced their paper and its ability to dissect the binding of splice site targeting drugs. The methods and supplemental text present an excellent resource for anyone trying to apply similar conceptual and experimental approaches to other biological systems. In general, the extra precision, clarity, and detail with the revision are much appreciated.

► We thank the referee for this favorable assessment, as well as for previous feedback, which played a major role in helping us improve this paper.

With that said, I have a few minor points for revision or clarification:

1. The biophysical models are completely thermodynamic under the assumption that U1 binding is strongly rate limiting for splicing. However, you argue from the empirical models that kinetic models may be required to explain splicing outcomes. I think the paper would benefit from a very brief consideration of how conclusions would or would not be altered in kinetic models where drugs can alter splicing rates, not just U1 occupancy.

► An in-depth analysis of specific kinetic models would be needed to properly address this point. We believe this is an interesting question for future research, but is too complex to answer in the current manuscript. We have added text after the discussion of kinetic proofreading to communicate this point. The revised text reads:

Although kinetic models of gene regulation can yield Hill coefficients larger than those of analogous thermodynamic models, specific mathematical models of kinetic proofreading in splicing would need to be analyzed to determine if such models can account for the anomalous cooperativity we observe. More generally, additional quantitative studies will be needed to establish the roles, if any, that feedback and proofreading play in sensitizing pre-mRNA transcripts to the effects of splice-modifying drugs.

2. Importantly, such considerations probably obviate the statements that the two-site model is “disproven”. It is very hard to disprove all models or know when all have been considered, and given the uncertainty in the biophysical models (Hill slopes $\neq 1$) I would caution against statements this strong.

► We have replaced all occurrences of “disprove” with “contradict”, one in the Abstract and one in the Introduction.

3. That said, the manuscript argues that their data argue against binding by two drug molecules. What would the dose-response curves look like with a Hill coefficient constrained to 2. How much worse is this at explaining the data, and what are the log-likelihoods of having integer vs.

floating Hill coefficients? This analysis would help strengthen the conclusion against multiple binding events.

► The Bayesian inference of Hill coefficients in our manuscript already provides the information the suggested analysis would produce, but in a more concise and interpretable way. In particular, the 95% credible intervals we obtain for the Hill coefficients show that all of these Hill coefficients (except for those in Fig. 5J and Fig. 6B,D) lie between 1 and 2 and differ from both 1 and 2 in a statistically significant manner. All of our code is available for users to reproduce our analysis, and to explore the full distribution of possible Hill coefficient values if they so choose. But we do not believe including additional analysis along these lines in the paper will help the readers understand our results.

4. Along the same lines, in lines 309-312, the authors hypothesize that in a dsRNA context, two molecules of drug could bind in flipped configurations, which would seem to contradict the conclusion that multiple binding does not occur. Could you please clarify the implications of such a model and whether it is consistent with your data?

► These conclusions are consistent. Our data support the conclusion that, for risdiplam, there is no second functional binding site within the interior of *SMN2* exon 7; rather, the only functional risdiplam binding site is in the 5'ss of *SMN2* exon 7. Our data also support the conclusion that branaplam (not risdiplam) can recognize its binding site in the 5'ss of *SMN2* exon 7 via two different interaction modes: one risdiplam-like interaction mode, and one hyper-activation interaction mode. Note that we are not claiming that two branaplam molecules can bind the 5'ss at the same time; we are claiming that a single branaplam molecule can bind in two different ways. To clarify this point, we have revised the first sentence of the paragraph to read:

We offer several possible structural explanations for the two putative modes of branaplam-dsRNA interaction at the 5'ss of *SMN2* exon 7.

The structural basis for the two hypothesized interaction modes of branaplam is unknown, but recognition of the 5'ss binding site in two flipped conformations is one possibility.

5. The discussion of synergy could still be clarified.

a. The choice of a linear mixture model was not immediately intuitive to this reviewer. The model proposed for synergy amounts to independent binding to multiple sites by multiple drugs. The total effect is then modeled as a linear combination of the distinct effects. However, it is not immediately clear how the molecular model connects to the parameters of the linear mixture model.

► The quadratic linear-mixture drug-effect model, used to analyze Fig. 6E-K, is not a mechanistic model. In particular, the parameters of this model do not correspond to specific physical quantities. Rather, this model empirically approximates the log of the inclusion/exclusion ratio using a quadratic function of drug mixture proportions. We do not use this model to come to any conclusions about the mechanisms of synergy; we use this model only to test whether there is any intermediate mixture that has higher activity than either individual drug. To clarify this point, we have revised the text to read:

P-values were computed using an empirical (i.e., non-mechanistic) Bayesian model described in SI Sec. 4.5.

b. The description of the different expectations for having synergy vs. no synergy could be made clearer by showing mock data plots or simulations for each case.

► We have revised the text to clarify this point. We believe the expectations under the synergy / no synergy hypotheses are now sufficiently described by this text, and that additional figures are not needed:

Under the no-synergy hypothesis, one should observe a monotonic response that interpolates between the two single-drug endpoints. On the other hand, if the inclusion/exclusion ratio is maximal at an intermediate mixture, this would indicate the presence of synergy between the two drugs.

c. More simply, different authors mean different things by “Synergy” so the authors should carefully define how they are using this term and define it physically and mathematically even if not simulated.

► We have clarified this point with the following text:

There are many different mathematical definitions for synergy⁶¹, but the operative definition for two-drug cocktails is whether a mixture of two drugs can yield a greater effect than either drug alone.

Ref. 61 is a review of the diverse mathematical criteria that have been used to define synergy.

Response to reviewer #3

Reviewer #3 (Remarks to the Author):

The authors have addressed the issues raised in my previous report and I recommend publication of the manuscript in Nature Communications.

Juan Valcarcel

► We thank the referee for this favorable assessment, as well as for previous feedback, which played a major role in helping us improve this paper.